# A Physics-Informed Machine Learning Framework for Safe and Optimal Control of Autonomous Systems

**Manan Tayal** [* 1]  **Aditya Singh** [* 1]  **Shishir Kolathaya** [1]  **Somil Bansal** [2]

## Abstract

As autonomous systems become more ubiquitous in daily life, ensuring high performance with guaranteed safety is crucial. However, safety and performance could be competing objectives, which makes their co-optimization difficult. Learning-based methods, such as Constrained Reinforcement Learning (CRL), achieve strong performance but lack formal safety guarantees due to safety being enforced as soft constraints, limiting their use in safety-critical settings. Conversely, formal methods such as Hamilton-Jacobi (HJ) Reachability Analysis and Control Barrier Functions (CBFs) provide rigorous safety assurances but often neglect performance, resulting in overly conservative controllers. To bridge this gap, we formulate the co-optimization of safety and performance as a state-constrained optimal control problem, where performance objectives are encoded via a cost function and safety requirements are imposed as state constraints. We demonstrate that the resultant value function satisfies a Hamilton-Jacobi-Bellman (HJB) equation, which we approximate efficiently using a novel physics-informed machine learning framework. In addition, we introduce a conformal prediction-based verification strategy to quantify the learning errors, recovering a high-confidence safety value function, along with a probabilistic error bound on performance degradation. Through several case studies, we demonstrate the efficacy of the proposed framework in enabling scalable learning of safe and performant controllers for complex, high-dimensional autonomous systems.

---

*Equal contribution [1]Center for Cyber Physical Systems, Indian Institute of Science, Bangalore, India [2]Department of Aeronautics and Astronautics, Stanford University, USA. Correspondence to: Manan Tayal <manantayal@iisc.ac.in>, Aditya Singh <adityasingh@iisc.ac.in>.

*Proceedings of the 42nd International Conference on Machine Learning*, Vancouver, Canada. PMLR 267, 2025. Copyright 2025 by the author(s).

## 1. Introduction

Autonomous systems are becoming increasingly prevalent across various domains, from self-driving vehicles and robotic automation to aerospace and industrial applications. Designing control algorithms for these systems involves balancing two fundamental objectives: *performance* and *safety*. Ensuring high performance is essential for achieving efficiency and task objectives under practical constraints, such as fuel limitations or time restrictions. For instance, a warehouse humanoid robot navigating to a destination must optimize its route for efficiency. At the same time, safety remains paramount to prevent catastrophic accidents or system failures. These two objectives, however, often conflict, making it challenging to develop control strategies that achieve both effectively.

A variety of data-driven approaches have been explored to integrate safety considerations into control synthesis. Constrained Reinforcement Learning (CRL) methods (Altman, 1999; Achiam et al., 2017) employ constrained optimization techniques to co-optimize safety and performance where performance is encoded as a reward function and safety is formulated as a constraint. These methods often incorporate safety constraints into the objective function, leading to only a soft imposition of the safety constraints. Moreover, such formulations typically minimize cumulative constraint violations rather than enforcing strict safety at all times, which can result in unsafe behaviors.

Another class of methods involves *safety filtering* (Hsu et al., 2024), which ensures constraint satisfaction by modifying control outputs in real-time. Methods such as Control Barrier Function (CBF)-based quadratic programs (QP) (Ames et al., 2017) and Hamilton-Jacobi (HJ) Reachability filters (Borquez et al., 2024; Wabersich et al., 2023) act as corrective layers on top of a (potentially unsafe) nominal controller, making minimal interventions to enforce safety constraints. However, because these safety filters operate independently of the underlying performance-driven controller, they often lead to myopic and suboptimal decisions. Alternatively, online optimization-based methods, such as Model Predictive Control (MPC) (García et al., 1989; Grüne et al., 2017) and Model Predictive Path Integral (MPPI) (Williams et al., 2018; Streichenberg et al., 2023), can naturally integrate

safety constraints while optimizing for a performance objective. These methods approximate infinite-horizon optimal control problems (OCPs) with a receding-horizon framework, enabling dynamic re-planning. While effective, solving constrained OCPs online remains computationally expensive, limiting their applicability for high-frequency control applications. The challenge is further exacerbated when dealing with nonlinear dynamics and nonconvex (safety) constraints, limiting the feasibility of these methods for ensuring safety and optimality for real-world systems.

A more rigorous approach to addressing the trade-off between performance and safety is to formulate the problem as a *state-constrained optimal control problem (SC-OCP)*, where safety is explicitly encoded as a hard constraint, while performance is expressed through a reward (or cost) function. While theoretically sound, characterizing the solutions of SC-OCPs is challenging unless certain controllability conditions hold (Soner, 1986). To address these challenges, (Altarovici et al., 2013) proposed an epigraph-based formulation, which characterizes the value function of an SC-OCP by computing its epigraph using dynamic programming, resulting in a Hamilton-Jacobi-Bellman Partial Differential Equation (HJB-PDE). The SC-OCP value function as well as the optimized policy are then recovered from this epigraph. However, dynamic programming suffers from the curse of dimensionality, making it impractical for high-dimensional systems with traditional numerical solvers (Mitchell, 2004; Wang et al., 2024). Furthermore, the epigraph formulation itself increases the problem's dimensionality, exacerbating computational complexity further. Many techniques for speeding up the computation of solutions to the HJB PDE put restrictions on the type of system allowed (Chow et al., 2017). However, solving the HJB PDE for general nonlinear systems remains a key challenge.

Recent advances in Deep Learning have enabled the development of physics-informed machine learning approaches (Raissi et al., 2017; 2019b) for solving partial differential equations (PDEs) with neural networks. These methods have demonstrated notable effectiveness in addressing high-dimensional PDEs while ensuring that the learned solutions adhere to the governing physical laws. In particular, DeepReach (Bansal & Tomlin, 2021) proposes a framework for solving Hamilton–Jacobi–Bellman (HJB) PDEs in safety-critical settings using physics-informed machine learning. However, its exclusive focus on safety neglects performance considerations, resulting in overly conservative control strategies.

In this work, we propose a novel algorithmic approach to *co-optimize safety and performance for high-dimensional autonomous systems*. Specifically, we formulate the problem as an SC-OCP and leverage the epigraph formulation in (Altarovici et al., 2013). To efficiently solve this epigraph

formulation, we leverage physics-informed machine learning (Raissi et al., 2019a; Li et al., 2022) to learn a solution to the resultant HJB-PDE by minimizing PDE residuals. This enables us to efficiently scale epigraph computation for higher-dimensional autonomous systems, leading to safe and performant policies. To summarize, our main contributions are as follows:

- We propose a novel Physics-Informed Machine Learning (PIML) framework to learn policies that co-optimize safety and performance for high-dimensional autonomous systems.

- We introduce a conformal prediction-based safety verification strategy that provides high-confidence probabilistic safety guarantees for the learned policy, reducing the impact of learning errors on safety.

- We propose a performance quantification framework that leverages conformal prediction to provide high-confidence probabilistic error bounds on performance degradation.

- Across three case studies, we showcase the effectiveness of our proposed method in jointly optimizing safety and performance, while scaling to complex, high-dimensional systems.

## 2. Problem Setup

Consider a nonlinear dynamical system characterized by the state $x \in \mathcal{X} \subseteq \mathbb{R}^n$ and control input $u \in \mathcal{U} \subseteq \mathbb{R}^m$, governed by the dynamics $\dot{x}(t) = f(x(t), u(t))$, where the function $f : \mathbb{R}^n \times \mathbb{R}^m \to \mathbb{R}^n$ is locally Lipschitz continuous. In this work, we assume that the dynamics model $f$ is known; however, it can also be learned from data if unavailable.

We are given a failure set $\mathcal{F} \subseteq \mathcal{X}$ that represents the set of unsafe states for the system (e.g., obstacles for an autonomous ground robot). The system's performance is quantified by the cost function $C(t, x, \mathbf{u})$, given by:

$$C(t, x(t), \mathbf{u}) = \int_{s=t}^{T} l(x(s)) \, ds + \phi(x(T)), \qquad (1)$$

where $l : \mathcal{X} \to \mathbb{R}_{\geq 0}$ and $\phi : \mathcal{X} \to \mathbb{R}_{\geq 0}$ are Lipschitz continuous and non-negative functions, representing the running cost over the time horizon $[t, T]$ and the terminal cost at time $T$, respectively. $\mathbf{u} : [t, T] \to \mathcal{U}$ is the control signal applied to the system. Using this premise, we define the main objective of this paper:

***Objective* 1.** We aim to synthesize an optimal policy $\pi^* : [t, T] \times \mathcal{X} \to \mathcal{U}$ that minimizes the cost function $C$ while ensuring that the system remains outside the failure set $\mathcal{F}$ at all times.

## 2.1. State-Constrained Optimal Control Problem

To achieve the stated objective, the first step is to encode the safety constraint via a function $g : \mathbb{R}^n \to \mathbb{R}$ such that, $\mathcal{F} := \{x \in \mathcal{X} \mid g(x) > 0\}$. Using these notations, the objective can be formulated as the following State-Constrained Optimal Control Problem (SC-OCP) to compute the value function $V$:

$$
\begin{aligned}
V(t, x(t)) = \min_{\mathbf{u}} \int_t^T & l(x(s))ds + \phi(x(T)) \\
\text{s.t. } & \dot{x} = f(x, u), \\
& g(x(s)) \leq 0 \quad \forall s \in [t, T]
\end{aligned}
\tag{2}
$$

This SC-OCP enhances the system's performance by minimizing the cost, while maintaining system safety through the state constraint, $g(x) \leq 0$, ensuring that the system avoids the failure set, $\mathcal{F}$. Thus, the policy, $\pi^*$, derived from the solution of this SC-OCP co-optimizes safety and performance.

## 2.2. Epigraph Reformulation

Directly solving the SC-OCP in (2) presents significant challenges due to the presence of (hard) state constraints. To address this issue, we reformulate the problem in its epigraph form (Boyd & Vandenberghe, 2004), which transforms the constrained optimization into a more tractable two-stage optimization problem. This reformulation allows us to efficiently obtain a solution to the SC-OCP in (2). The resulting formulation is given by:

$$
\begin{aligned}
V(t, x(t)) = \min_{z \in \mathbb{R}^+} & \ z \\
\text{s.t. } & \hat{V}(t, x, z) \leq 0,
\end{aligned}
\tag{3}
$$

where $z$ is a non-negative auxiliary optimization variable, and $\hat{V}$ represents the auxiliary value function. Here, $\hat{V}$ is defined as (Altarovici et al., 2013):

$$
\hat{V}(t, x(t), z) = \min_{\mathbf{u}} \max\{C(t, x(t), \mathbf{u}) - z, \max_{s \in [t, T]} g(x(s))\}.
\tag{4}
$$

Note that if $\hat{V}(t, x, z) < 0$, it implies that $g(x(s)) < 0$ for all $s \in [t, T]$. In other words, the system must be outside the failure set at all times; therefore, the system is guaranteed to be safe whenever $\hat{V}(t, x, z) < 0$.

In this reformulated problem, state constraints are effectively eliminated, enabling the use of dynamic programming to characterize the value function, as we explain later in this section. Intuitively, optimal $z$ ($z^*$) can be thought of as the *minimum permissible cost* the policy can incur without compromising on safety. From Equation 3, it can be inferred that if $z > z^*$, the safety constraint dominates in the max term, resulting in a conservative policy. Conversely, if $z < z^*$, the performance objective takes precedence, leading to a potentially aggressive policy that might compromise safety.

Furthermore, to facilitate solving the epigraph reformulation, $z$ can be treated as a state variable, with its dynamics given by $\dot{z}(t) = -l(x(t))$. This implies that as the trajectory progresses over time, the minimum permissible cost, $z$, decreases by the step cost $l(x)$ at each time step. This allows us to define an augmented system that evolves according to the following dynamics:

$$
\dot{\hat{x}} = \hat{f}(t, \hat{x}, u) := \begin{bmatrix} f(t, x, u) \\ -l(x) \end{bmatrix},
\tag{5}
$$

where $\hat{x} := [x, z]^T$ represents the augmented state. With the augmented state representation and under assumptions A1–A4 of (Altarovici et al., 2013), the auxiliary value function $\hat{V}(t, x(t), z(t))$ is a unique continuous viscosity solution satisfying the following Hamilton–Jacobi–Bellman (HJB) PDE:

$$
\min\left(-\partial_t \hat{V} - \min_{\mathbf{u}}\langle \nabla_{\hat{x}} \hat{V}(t, \hat{x}), \hat{f}(\hat{x}, u)\rangle, \hat{V} - g(x)\right) = 0,
\tag{6}
$$

$\forall t \in [0, T)$ and $\hat{x} \in \mathcal{X} \times \mathbb{R}$, where $\langle \cdot, \cdot \rangle$ denotes the dot product of vectors. The boundary condition for the PDE is given by:

$$
\hat{V}(T, \hat{x}) = \max\left(\phi(x(T)) - z, g(x)\right), \quad \hat{x} \in \mathcal{X} \times \mathbb{R}.
\tag{7}
$$

Note that by a slight abuse of notations, we have replaced the arguments $x, z$ for $\hat{V}$ with the augmented state $\hat{x}$.

# 3. Methodology

To solve the SC-OCP in Equation (2), we aim to compute the optimal value function $V$, which minimizes the cost while ensuring system safety. In this section, we outline a structured approach: first, we learn the auxiliary value function $\hat{V}$ using a physics-informed machine learning framework. Then, we apply a conformal prediction-based method to verify safety and correct for potential learning errors in $\hat{V}$. The final value function $V$ is obtained from the safety-corrected $\hat{V}$ using the epigraph formulation in (3). Lastly, we assess the performance of $V$ through a second conformal prediction procedure. Figure 1 gives an overview of the proposed approach. The following subsections provide a detailed explanation of each step, beginning with the methodology for learning $\hat{V}$.

## 3.1. Training the Auxiliary Value Function ($\hat{V}$)

The auxiliary value function, $\hat{V}$, satisfies the HJB-PDE in Equation (6), as discussed in Section 2.2. Traditionally, numerical methods are used to solve the HJB-PDE over a grid representation of the state space (Mitchell, 2004; Schmerling, 2021), where time and spatial derivatives are approximated numerically. While grid-based methods are accurate for low-dimensional problems, they struggle with

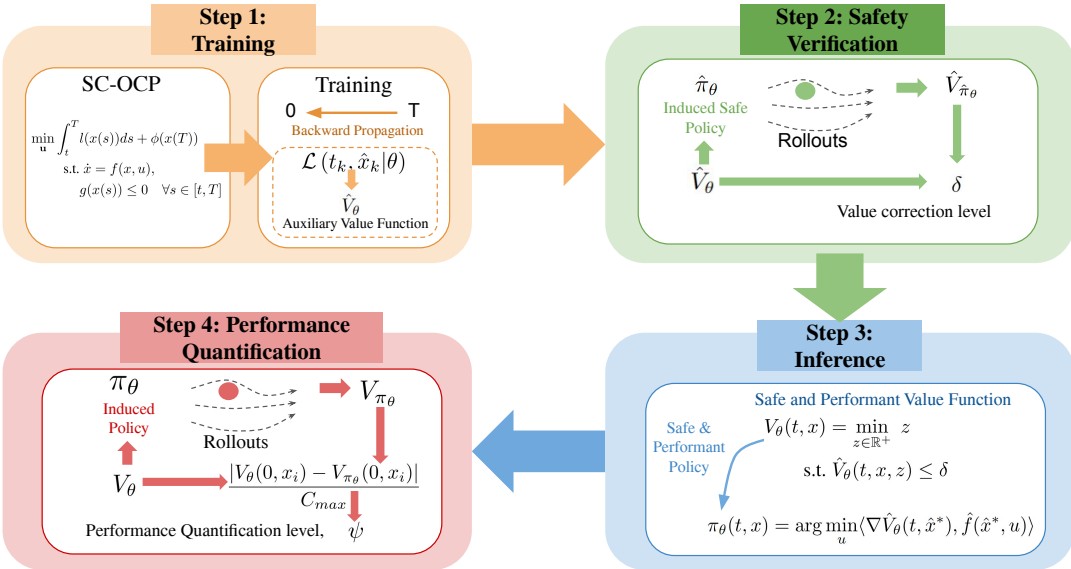

*Figure 1.* **Overview of the proposed approach**: The methodology is organized into four steps. The **first step** involves training the auxiliary value function, $\hat{V}_\theta$, using a physics-informed machine learning framework. The **second step** applies a conformal prediction approach for safety verification of the learned $\hat{V}_\theta$. In the **third step**, the final value function $V_\theta$ and the optimal safe and performant policy $\pi_\theta$ are inferred. The **fourth step** quantifies the performance of $V_\theta$ through a second conformal prediction procedure.

the curse of dimensionality – their computational complexity increases exponentially with the number of states – limiting their use in high-dimensional systems. To address this, we adopt a physics-informed machine learning framework, inspired by (Bansal & Tomlin, 2021), which has proven effective for high-dimensional reachability problems.

The solution of the HJB-PDE inherently evolves backward in time, as the value function at time $t$ is determined by its value at $t + \Delta t$. To facilitate neural network training, we use a curriculum learning strategy, progressively expanding the time sampling interval from the terminal time $[T, T]$ to the full time horizon $[0, T]$. This approach allows the neural network to first accurately learn the value function from the terminal boundary conditions, subsequently propagating the solution backward in time by leveraging the structure of the HJB-PDE.

Specifically, the auxiliary value function is approximated by a neural network, $\hat{V}_\theta$, where $\theta$ denotes the trainable parameters of the network. Training samples, $(t_k, x_k, z_k)_{k=1}^N$, are randomly drawn from the state space based on the curriculum training scheme. The proposed learning framework utilizes a loss function that enforces two primary objectives: (i) compliance with the PDE in (6), using the PDE residual error given by:

$$\mathcal{L}_{pde}(t_k, \hat{x}_k|\theta) = \| \min \left\{ -\partial_t \hat{V}_\theta(t_k, \hat{x}_k) - H(t_k, \hat{x}_k), \right.$$
$$\left. \hat{V}_\theta(t_k, \hat{x}_k) - g(x_k) \right\} \|, \tag{8}$$

where $H(t, \hat{x}) = \min_{u \in \mathcal{U}} \langle \nabla \hat{V}_\theta(t, \hat{x}), \hat{f}(\hat{x}, u) \rangle$ and (ii) satisfaction of the boundary condition in (7), using boundary condition loss, given by:

$$\mathcal{L}_{bc}(t_k, \hat{x}_k|\theta) = \| \max \left( \phi(x_k) - z_k, g(x_k) \right) -$$
$$\hat{V}_\theta(t_k, \hat{x}_k) \| \, \mathbb{1}(t_k = T). \tag{9}$$

These terms are balanced by a trade-off parameter $\lambda$, leading to the overall loss function:

$$\mathcal{L}(t_k, \hat{x}_k|\theta) = \mathcal{L}_{pde}(t_k, \hat{x}_k|\theta) + \lambda \mathcal{L}_{bc}(t_k, \hat{x}_k|\theta) \tag{10}$$

Furthermore, we use the adaptive loss re-balancing scheme proposed in (Wang et al., 2021) to reduce the impact of $\lambda$ on the learned value function. Minimizing the overall loss function provides a self-supervised learning mechanism to approximate the auxiliary value function.

### 3.2. Safety Verification

The learned auxiliary value function, $\hat{V}_\theta$, induces a policy, $\hat{\pi}_\theta$, that minimizes the Hamiltonian term $H(t, \hat{x})$ in the HJB-PDE. The policy is given by:

$$\hat{\pi}_\theta(t, \hat{x}) = \arg\min_{u \in \mathcal{U}} \langle \nabla \hat{V}_\theta(t, \hat{x}), \hat{f}(\hat{x}, u) \rangle. \tag{11}$$

The rollout cost corresponding to this policy is defined as:

$$\hat{V}_{\hat{\pi}_\theta}(t, \hat{x}) = \max\{C(t, x(t), \mathbf{u}) - z, \max_{s \in [t, T]} g(x(s))\}\Big|_{\mathbf{u} = \hat{\pi}_\theta} \tag{12}$$

Ideally, the rollout cost from a given state under $\hat{\pi}_\theta$ should match the value of the auxiliary value function at that state.

**Algorithm 1** Safety Verification using Conformal Prediction

**Require:** $\mathcal{S}$, $N_s$, $\beta_s$, $\epsilon_s$, $\hat{V}_\theta(\hat{x}, 0)$, $\hat{V}_{\hat{\pi}_\theta}(\hat{x}, 0)$, $M$ (number of $\delta$-levels to search for $\delta$),
1: $D_0 \leftarrow$ Sample $N_s$ IID states from $\mathcal{S}_{\delta=0}$
2: $\delta_0 \leftarrow \min_{\hat{x}_j \in D_0}\{\hat{V}_\theta(0, \hat{x}_j) : \hat{V}_{\hat{\pi}_\theta}(0, \hat{x}_j) \geq 0\}$
3: $\epsilon_0 \leftarrow$ (14) (using $\alpha_{\delta=0}$)
4: $\Delta \leftarrow$ Ordered list of $M$ uniform samples from $[\delta_0, 0]$
5: **for** $i = 0, 1, \ldots, M - 1$ **do**
6:    **while** $\epsilon_i \leq \epsilon_s$ **do**
7:       $\delta_i \leftarrow \Delta_i$
8:       Update $\alpha_{\delta_i}$ from $\delta_i$
9:       $\epsilon_i \leftarrow$ (14) (using $\alpha_{\delta_i}$)
10:    **end while**
11: **end for**
12: **return** $\delta \leftarrow \delta_i$

**Algorithm 2** Performance Quantification using Conformal Prediction

**Require:** $\mathcal{S}^*$, $N_p$, $\beta_p$, $V_\theta(x, 0)$, $V_{\pi_\theta}(x, 0)$
1: $D \leftarrow$ Sample $N_p$ IID states from$\{x : x \in \mathcal{S}^*\}$
2: **for** $i = 0, 1, \ldots, N_p - 1$ **do**
3:    $P_i \leftarrow p_i(0, D)$
4: **end for**
5: $P \leftarrow P$ sorted in decreasing order
6: $\alpha_p \leftarrow \frac{1}{N_p+1}$, $\psi_0 \leftarrow P_0$, $\epsilon_0 \leftarrow$ (18)
7: **for** $i = 0, 1, \ldots, N_p - 1$ **do**
8:    **while** $\epsilon_i \leq \epsilon_p$ **do**
9:       $\alpha_p \leftarrow \frac{i+1}{N_p+1}$, $\psi_i \leftarrow P_i$, $\epsilon_i \leftarrow$ (18)
10:    **end while**
11: **end for**
12: **return** $\psi \leftarrow \psi_i$

However, due to learning inaccuracies, discrepancies can arise. This becomes critical when a state, $\hat{x}_i$, is deemed safe by the auxiliary value function ($\hat{V}_\theta(t, \hat{x}) \leq 0$) but is unsafe under the induced policy ($\hat{V}_{\hat{\pi}_\theta}(t, \hat{x}) > 0$). To address this, we introduce a uniform value function correction margin, $\delta$, which guarantees that the sub-$\delta$ level set of the auxiliary value function remains safe under the induced policy. Mathematically, the optimal $\delta$ ($\delta^*$) can be expressed as:

$$\delta^* := \min_{\hat{x} \in \mathcal{X}}\{\hat{V}_\theta(0, \hat{x}) : \hat{V}_{\hat{\pi}_\theta}(0, \hat{x}) \geq 0\} \tag{13}$$

Intuitively, $\delta^*$ identifies the tightest level of the value function that separates safe states under $\hat{\pi}_\theta$ from unsafe ones. Hence, any initial state within the sub-$\delta^*$ level set is guaranteed to be safe under the induced policy, $\hat{\pi}_\theta^*$. However, calculating $\delta^*$ exactly requires infinitely many state-space points. To overcome this, we adopt a conformal-prediction-based approach to approximate $\delta^*$ using a finite number of samples, providing a probabilistic safety guarantee. The following theorem formalizes our approach:

**Theorem 3.1** (Safety Verification Using Conformal Prediction). *Let $\mathcal{S}_\delta$ be the set of states satisfying $\hat{V}_\theta(0, \hat{x}) \leq \delta$, and let $(0, \hat{x}_i)_{i=1,\ldots,N_s}$ be $N_s$ i.i.d. samples from $\mathcal{S}_\delta$. Define $\alpha_\delta$ as the safety error rate among these $N_s$ samples for a given $\delta$ level. Select a safety violation parameter $\epsilon_s \in (0, 1)$ and a confidence parameter $\beta_s \in (0, 1)$ such that:*

$$\sum_{i=0}^{l-1} \binom{N_s}{i} \epsilon_s^i (1 - \epsilon_s)^{N_s - i} \leq \beta_s, \tag{14}$$

*where $l = \lfloor (N_s + 1)\alpha_\delta \rfloor$. Then, with the probability of at least $1 - \beta_s$, the following holds:*

$$\mathbb{P}_{\hat{x} \in \mathcal{S}_\delta}\left(\hat{V}(0, \hat{x}_i) \leq 0\right) \geq 1 - \epsilon_s. \tag{15}$$

The proof is available in Appendix A.1. The safety error rate $\alpha_\delta$ is defined as the fraction of samples satisfying $\hat{V}_\theta \leq \delta$ and $\hat{V}_{\hat{\pi}_\theta} \geq 0$ out of the total $N_s$ samples.

Algorithm 1 presents the steps to calculate $\delta$ using the approach proposed in this theorem.

### 3.3. Obtaining Safe and Performant Value Function and Policy from $\hat{V}_\theta$

Using the $\delta$-level estimate from Algorithm (1), we can finally obtain the safe and performant value function, $V_\theta(t, x)$, by solving the following epigraph optimization problem:

$$V_\theta(t, x) = \min_{z \in \mathbb{R}^+} z$$
$$\text{s.t. } \hat{V}_\theta(t, x, z) \leq \delta. \tag{16}$$

Note that $V_\theta(t, x)$ is trivially $\infty$ for the states where $\hat{V}_\theta(t, x, z) > \delta$, since such states are unsafe and hence do not satisfy the safety constraint.

In practice, we solve this optimization problem by using a binary search approach on $z$. The resulting optimal state-feedback control policy, $\pi_\theta : \mathcal{X} \times [t, T] \rightarrow \mathcal{U}$, satisfying Objective (1), is given by:

$$\pi_\theta(t, x) = \arg\min_u \langle \nabla \hat{V}_\theta(t, \hat{x}^*), \hat{f}(\hat{x}^*, u) \rangle, \tag{17}$$

where $\hat{x}^*$ is the augmented state associated with the optimal $z^*$ obtained by solving (16), i.e., $\hat{x}^* = [x, z^*]^T$. Intuitively, we can expect $\pi_\theta$ to learn behaviors that best tradeoff the safety and performance of the system.

### 3.4. Performance Quantification

In general, the learning inaccuracies in the auxiliary value function $\hat{V}_\theta$, may lead to errors in the value function $V_\theta$. These errors, in turn, can lead to performance degradation under policy $\pi_\theta$.

To quantify this degradation, we propose a conformal prediction-based performance quantification method that provides a probabilistic upper bound on the error between

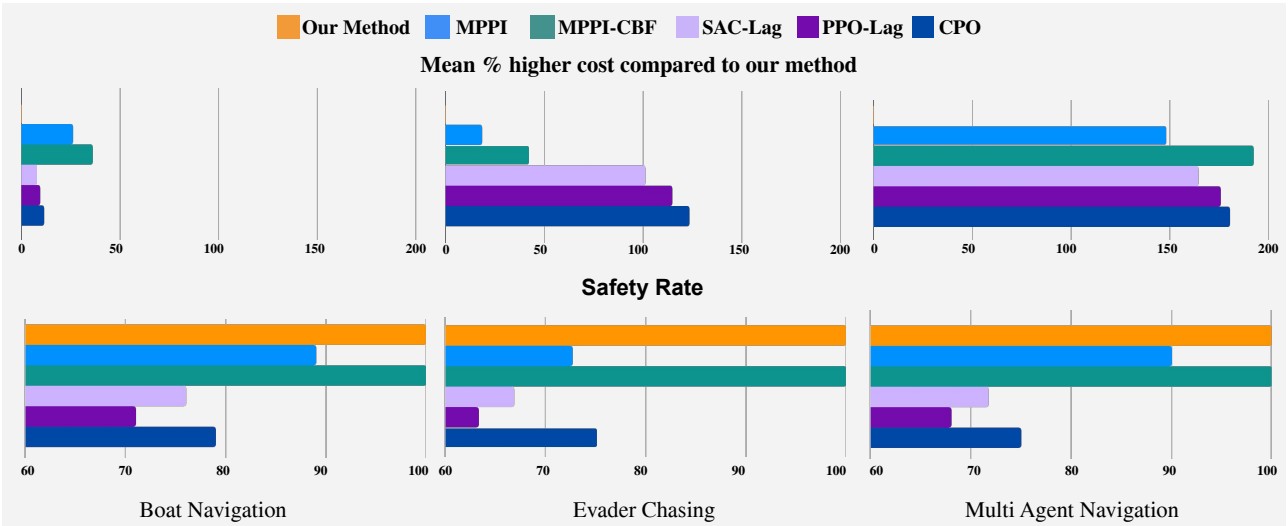

*Figure 2.* This figure presents a comparative study between all the methods based on our evaluation metrics. The top plot illustrates the **mean percentage increase in cumulative cost** relative to our method for each baseline, demonstrating that our approach consistently incurs lower costs, with the gap widening as system complexity grows. The bottom plot depicts the **safety rates**, showing that our method maintains a 100% safety rate, while baselines that encourage safety rather than enforcing it (like MPPI and C-SAC) achieve lower rates. MPPI-CBF also attains 100% safety but at the expense of performance. Overall, our method uniquely **balances both safety and performance**, whereas the baselines compromise on at least one aspect.

the value function and the value obtained from the induced policy. The following theorem formalizes our approach:

**Theorem 3.2** (Performance Quantification Using Conformal Prediction). *Suppose $\mathcal{S}^*$ denotes the safe states satisfying $V_\theta(0, x) < \infty$ (or equivalently $\hat{V}_\theta(0, \hat{x}^*) < \delta$) and $(0, x_i)_{i=1,\dots,N_p}$ are $N_p$ i.i.d. samples from $\mathcal{S}^*$. For a user-specified level $\alpha_p$, let $\psi$ be the $\frac{\lceil (N_p+1)(1-\alpha_p) \rceil}{N_p}$th quantile of the scores $(p_i := \frac{|V_\theta(0,x_i) - V_{\pi_\theta}(0,x_i)|}{C_{max}})_{i=1,\dots,N_p}$ on the $N_p$ state samples. Select a violation parameter $\epsilon_p \in (0, 1)$ and a confidence parameter $\beta_p \in (0, 1)$ such that:*

$$\sum_{i=0}^{l-1} \binom{N_p}{i} \epsilon_p^i (1 - \epsilon_p)^{N_p - i} \leq \beta_p \quad (18)$$

*where, $l = \lfloor (N_p + 1)\alpha_p \rfloor$. Then, the following holds, with probability $1 - \beta_p$:*

$$\mathbb{P}_{x \in \mathcal{S}^*} \left( \frac{|V_\theta(0, x_i) - V_{\pi_\theta}(0, x_i)|}{C_{max}} \leq \psi \right) \geq 1 - \epsilon_p. \quad (19)$$

*where $C_{max}$ is a normalizing factor and denotes the maximum possible cost that could be incurred for any $x \in \mathcal{S}^*$.*

The proof is available in Appendix A.2. Note that $C_{max}$ can be easily calculated by calculating the upper bound of the cost function $C(t, x(t), \mathbf{u}) \forall x \in \mathcal{S}^*$.

Intuitively, the performance of the resultant policy is the best when the $\psi$ value approaches 0, while the worst performance occurs at $\psi = 1$. Algorithm 2 presents the steps to calculate $\psi$ using the approach proposed in this theorem.

## 4. Experiments

The objective of this paper is to demonstrate the co-optimization of performance and safety. To achieve this, we evaluate the proposed method and compare them with baselines using three metrics: (1) **Cumulative Cost:** This metric represents the total cost $\int_0^T l(x(s))ds + \phi(x(T))$, accumulated by a policy over the safe trajectories. (2) **Safety Rate:** This metric is defined as the percentage of trajectories that remain safe, i.e., never enter the failure region $\mathcal{F}$ at any point in time. (3) **Computation Time:** This metric compares the offline and online computation times of our method and the baselines.

**Baselines**: We consider two categories of baselines: the first set of methods aims to enhance the system performance (i.e., minimize the cumulative cost) while encouraging safety, encompassing methods such as Lagrangian-based CRL algorithms like SAC-Lagrangian (SAC-Lag), PPO-Lagrangian (PPO-Lag) (Ray et al., 2019; Ji et al., 2024) and Model Predictive Path Integral (MPPI) (Williams et al., 2018) algorithms. The second category prioritizes safety, potentially at the cost of performance. This includes Constrained Policy Optimization (CPO) (Achiam et al., 2017) and safety filtering techniques such as Control Barrier Function (CBF)-based quadratic programs (QP) (Ames et al., 2017) that modify a nominal, potentially unsafe controller to satisfy the safety constraint.

### 4.1. Efficient and Safe Boat Navigation

In our first experiment, we consider a 2D autonomous boat navigation problem, where a boat with coordinates $(x_b, y_b)$

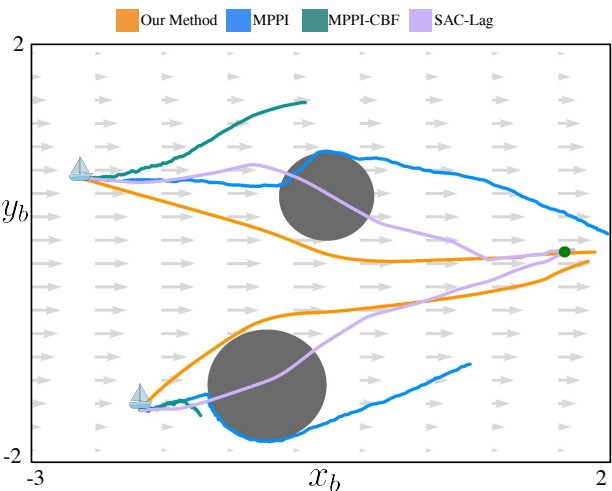

*Figure 3.* Trajectories from two distinct initial states are shown, with dark grey circles representing obstacles and the green dot indicating the goal at $[1.5, 0]^T$. Notably, our method is the **only one** that **successfully approaches the goal** while **adhering to safety constraints**.

navigates a river with state-dependent drift to reach an island. The boat must avoid two circular boulders (obstacles) of different radii, which corresponds to the safety constraint in the system (see Fig. 3). The cost function penalizes the distance to the goal. The system state, $x$, evolves according to the dynamics:
$$x = [x_b, y_b], \quad \dot{x} = [u_1 + 2 - 0.5y_b^2, u_2] \quad (20)$$

where $[u_1, u_2]$ are the bounded control inputs in the $x_b$ and $y_b$ directions, constrained by the control space $\mathcal{U} = \{[u_1, u_2] \in \mathbb{R}^2 \mid \|[u_1, u_2]\| \leq 1\}$. The term $2 - 0.5y_b^2$ introduces a state-dependent drift, complicating the control task as the actions must counteract the drift while ensuring safety, which is challenging under bounded control inputs. The rest of the details about the experiment setup can be found in the Appendix B.1.

**Safety Guarantees and Performance Quantification**: We use $N_s = 300K$ and $N_p = 300K$ samples for thorough verification, ensuring dense state space sampling. For this experiment, we set $\epsilon_s = 0.001$ and $\beta_s = 10^{-10}$, resulting in a $\delta$-level of 0. This implies that, with $1 - 10^{-10}$ confidence, any state with $\hat{V}_\theta(t, x, z) \leq 0$, is safe with at least 99.9% probability. For performance quantification, we set $\epsilon_p = 0.01$ and $\beta_p = 10^{-10}$, leading to a $\psi$-level of 0.136. This ensures, with $1 - 10^{-10}$ confidence, that any state in $\mathcal{S}^*$ has a normalized error between the predicted value and the policy value of less than 0.136 with 99% probability. Low $\delta$ and $\psi$ values with high confidence indicate that the learned policy closely approximates the optimal policy and successfully co-optimizes safety and performance.

**Baselines**: This being a 2-dimensional system, we compare our method with the ground truth value function computed by solving the HJB-PDE numerically using the Level

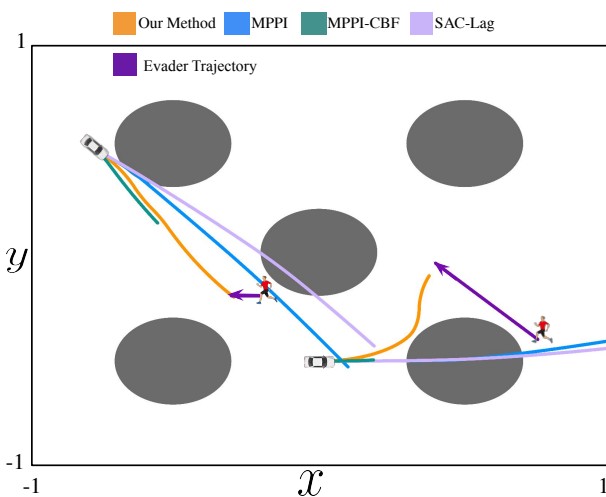

*Figure 4.* Trajectories from two distinct initial states are depicted, with dark grey circles representing obstacles and purple trajectories indicating the evader's path, with arrows showing its direction of motion. Our method successfully **tracks the evader** while **avoiding collisions**, whereas all other methods either fail to maintain safety, struggle to track the evader or both

Set Toolbox (Mitchell, 2004) (results in Appendix B.1.1). Additional baselines include: (1) MPPI, a sample-based path-planning algorithm with safety as soft constraints, (2) MPPI-NCBF, where safety is enforced using a Neural CBF-based QP with MPPI as the nominal controller (Dawson et al., 2022; Tayal et al., 2024b), and (3) Constrained RL methods like SAC-Lag, PPO-Lag, and CPO.

**Comparative Analysis:** Figure 3 shows that our method effectively reaches the goal while avoiding obstacles, even when starting close to them. In contrast, MPPI and CRL-based policies fail to maintain safety, while MPPI-NCBF ensures safety but performs poorly (leading to very slow trajectories). Figure 2 highlights that our method outperforms all others. SAC-Lag attains a mean cost that is 7.5% higher than ours, while exhibiting the lowest safety rate at 76%. The remaining CRL methods display comparable trends, highlighting their inability to jointly optimize for safety and performance. MPPI, with a more competitive safety rate of 89%, performs poorly with a 32.67% higher mean cost. MPPI-NCBF achieves 100% safety but performs significantly worse, with a 50.72% higher mean cost. Additionally, CBF-based controllers sometimes violate control bounds, limiting their applicability. This demonstrates that our method balances safety and performance, unlike others that compromise on one aspect. Moreover, the 100% safety rate of our method aligns closely with at least 99.9% safety level that we expect using our proposed verification strategy, providing empirical validation of the safety assurances.

### 4.2. Pursuer Vehicle tracking a moving Evader

In our second experiment, we consider an acceleration-driven pursuer vehicle, tracking a moving evader while

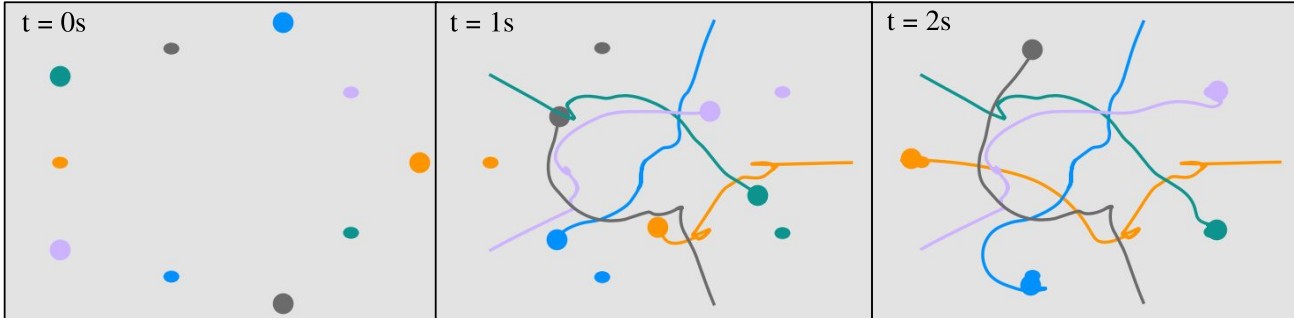

*Figure 5.* Snapshots of multi-agent navigation trajectories at different times using the proposed method. Agents are represented as circles with radius $R$, indicating the minimum safe distance they must maintain from each other. Smaller dots mark their respective goals. The trajectories show that agents proactively **maintain long-horizon safety** by adjusting their paths to avoid close encounters, rather than enforcing safety reactively, which could lead to suboptimal behaviors. Finally, the agents **reach their respective goals within the specified time horizon**.

avoiding five circular obstacles (see Fig. 4). This experiment involves an 8-dimensional system, with the state $x$ defined as $x = [x_p, y_p, v, \Theta, x_e, y_e, v_{xe}, v_{ye}]^T$, where $x_p, y_p, v, \Theta$ represent the coordinates, linear velocity, and orientation of the pursuer vehicle, respectively, and $x_e, y_e, v_{xe}, v_{ye}$ represent the coordinates and linear velocities of the evader vehicle. The pursuer vehicle is controlled by linear acceleration ($u_1$) and angular velocity ($u_2$). The control space is $\mathcal{U} = \{[u_1, u_2] \in [-2, 2]^2\}$. The complexity of this system stems from the dynamic nature of the goal, along with the challenge of ensuring safety in a cluttered environment, which in itself is a difficult safety problem. More details about the experiment setup are in Appendix B.2.

**Safety Guarantees and Performance Quantification**: Similar to the previous experiment, we set $N_s = N_p = 300k$. We choose $\epsilon_s = 0.01$ and $\beta_s = 10^{-10}$, yielding a $\delta$-level of $-0.04$ and a safety level of $99\%$ on the auxiliary value function. For performance, we set $\epsilon_p = 0.01$ and $\beta_p = 10^{-10}$, leading to a $\psi$-level of $0.137$. These values indicate the learned policy maintains high safety with low-performance degradation in this cluttered environment.

**Baselines:** As in the previous experiment, we employ MPPI and CRL methods (SAC-Lag, PPO-Lag, and CPO). For safety filtering, we utilize a QP based on the collision cone CBF (C3BF) (Goswami et al., 2024), chosen for its effectiveness in managing acceleration-driven systems.

**Comparative Analysis:** Figure 4 shows that our method effectively tracks the moving evader while avoiding obstacles, even when starting close to them. In contrast, other methods have limitations: MPPI and CRL methods attempt to follow the evader but fail to maintain their pace, violating safety constraints, while MPPI-C3BF sacrifices performance to maintain safety. Figure 2 highlights our method's superior performance in balancing safety and performance. MPPI achieves the best performance among the baselines but with an 18% higher mean cost and only a 72% safety rate. MPPI-

NCBF ensures 100% safety but has a 42% higher mean cost. SAC-Lag underperforms both in safety (66% safety rate) and performance (101% higher mean cost). A similar trend is evident across all other CRL methods, indicating their difficulty in co-optimizing safety and performance in high-dimensional, complex systems.

### 4.3. Multi-Agent Navigation

In our third experiment, we consider a multi-agent setting where each of the 5 agents, represented by $x_i = [x_{a_i}, y_{a_i}, x_{g_i}, y_{g_i}]$, tries to reach its goal while avoiding collisions with others. $(x_{a_i}, y_{a_i})$ denote the position of the $i$th agent, while $(x_{g_i}, y_{g_i})$ represent the goal locations for that agent. The system is 20-dimensional, with each agent controlled by its $x$ and $y$ velocities. The control space for each agent is $\mathcal{U}_i = \{[v_{x_i}, v_{y_i}] \mid \|[v_{x_i}, v_{y_i}]\| \leq 1\}$. The complexity of this system stems from the interactions and potential conflicts between agents as they attempt to reach their goals while avoiding collisions. The rest of the details about the experiment setup can be found in Appendix B.3.

**Safety Guarantees and Performance Quantification**: We set $N_s = N_p = 300k$, $\epsilon_s = 0.001$, and $\beta_s = 10^{-10}$, resulting in a $\delta$-level of $-0.09$ with safety assurance of $99.9\%$ for the auxiliary value function. For performance quantification, we set $\epsilon_p = 0.01$ and $\beta_p = 10^{-10}$, leading to a $\psi$-level of $0.068$. It is evident that the $\delta$ and $\psi$ values remain very low with high confidence, highlighting the effectiveness of our method in co-optimizing safety and performance for high-dimensional, multi-agent systems.

**Baselines:** Similar to previous experiments, we have used MPPI, SAC-Lag, PPO-Lag, CPO, and MPPI-NCBF as our baselines for this experiment too.

**Comparative Analysis:** Figure 5 shows that our method ensures long-horizon safety while enabling all agents to reach their goals without collisions. In contrast, the baseline methods either exhibit overly conservative behavior or

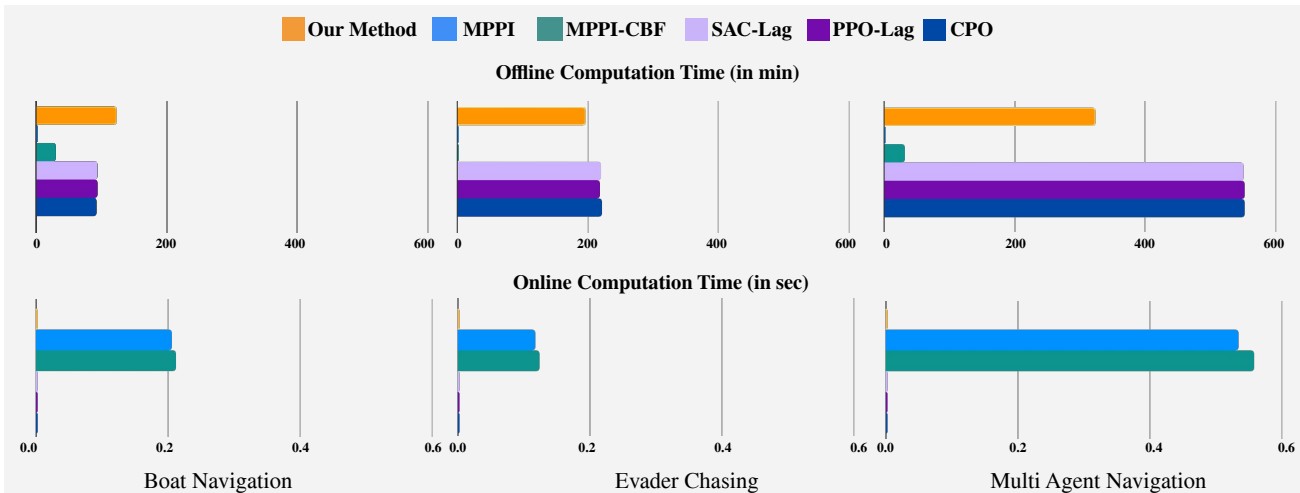

*Figure 6.* This figure presents a comparative analysis of all methods based on online and offline computation time evaluated on the same computing machine. The top plot illustrates the **offline computation time** for our method and the baselines. Since our method and SAC-Lag involve training value functions, they incur higher offline computation costs, whereas MPPI-based methods require no offline training. The bottom plot depicts the **online computation time**, demonstrating that our method and SAC-Lag have minimal online computation requirements, whereas MPPI-based methods exhibit significantly higher online computational costs.

fail to maintain safety, leading to collisions, as detailed in Appendix B.3.1. Figure 2 demonstrates the superior performance of our approach, with MPPI, MPPI-NCBF, and SAC-Lag showing mean percentage cost increases of 148%, 192%, and 164%, respectively. Although MPPI and MPPI-NCBF achieve competitive safety rates of 90% and 100%, their significant performance degradation highlights their inability to balance safety and performance in complex systems. MPPI's subpar performance stems from its reliance on locally optimal solutions in a finite data regime, leading to several deadlocks along the way and overall suboptimal trajectories over a long horizon. Furthermore, CRL methods struggle with both safety and performance, further demonstrating their limitations in handling increasing system complexity and dimensionality. These results confirm our method's ability to co-optimize safety and performance in high-dimensional systems, demonstrating its scalability. Additionally, the safety guarantees hold in the test samples, validating the scalability of our safety verification framework for multi-agent systems.

### 4.4. Computation time Analysis

Figure 6 presents a comparative analysis of the offline and online computation times for our method against the baselines. While traditional grid-based methods suffer from an exponentially scaling computational complexity (and are completely intractable for the 8D Evader Chasing and 20D Multi-Agent case studies), the proposed method scales much better with the system dimensionality. For example, the computation time increases only minimally from the 2D system to the 8D system, thanks to neural network parallelization. Similarly, the computation time increases

sublinearly from 8D to 20D system. This scalability is a key advantage of the proposed approach. We finally note that while offline training requires time, our method achieves real-time inference speeds, with optimal policy computed in just **2ms** across all systems, making the approach highly suitable for real robotic systems.

## 5. Conclusion and Future Work

In this work, we introduced a physics-informed machine learning framework for co-optimizing safety and performance in autonomous systems. By formulating the problem as a state-constrained optimal control problem (SC-OCP) and leveraging an epigraph-based approach, we enabled scalable computation of safety-aware policies. Our method integrates conformal prediction-based safety verification to ensure high-confidence safety guarantees while maintaining optimal performance. Through multiple case studies, we demonstrated the effectiveness and scalability of our approach in high-dimensional systems. In future, we will explore methods for rapid adaptation of the learned policies in light of new information about the system dynamics, environments, or safety constraints. We will also apply our method to other high-dimensional autonomous systems and systems with unknown dynamics.

## Acknowledgements

Manan is supported by the Prime Minister's Research Fellowship (PMRF), Government of India. This work is partially supported by the AI & Robotics Technology Park (ARTPARK) at IISc, the DARPA Assured Neuro Symbolic

Learning and Reasoning (ANSR) program, and the NSF CAREER award (2240163).

## Impact Statement

This paper presents an approach to co-optimize safety and performance in autonomous systems using Physics-Informed Machine Learning. This framework advances the deployment of scalable, provably safe, and high-performance controllers for complex, high-dimensional autonomous systems, with potential applications in robotics and autonomous driving.

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

# Contents

# A. Proofs

### A.1. Theorem (3.1)

**Theorem 3.1** (Safety Verification Using Conformal Prediction) Let $\mathcal{S}_\delta$ be the set of states satisfying $\hat{V}_\theta(0, \hat{x}) \leq \delta$, and let $(0, \hat{x}_i)_{i=1,\ldots,N_s}$ be $N_s$ i.i.d. samples from $\mathcal{S}_\delta$. Define $\alpha_\delta$ as the safety error rate among these $N_s$ samples for a given $\delta$ level. Select a safety violation parameter $\epsilon_s \in (0, 1)$ and a confidence parameter $\beta_s \in (0, 1)$ such that:

$$\sum_{i=0}^{l-1} \binom{N_s}{i} \epsilon_s^i (1 - \epsilon_s)^{N_s - i} \leq \beta_s,$$

where $l = \lfloor (N_s + 1)\alpha_\delta \rfloor$. Then, with the probability of at least $1 - \beta_s$, the following holds:

$$\mathbb{P}_{\hat{x} \in \mathcal{S}_\delta} \left( \hat{V}(0, \hat{x}_i) \leq 0 \right) \geq 1 - \epsilon_s.$$

*Proof.* Before we proceed with the proof of the Theorem (3.1), let us look at the following lemma which describes split conformal prediction:

*Lemma* 1 (Split Conformal Prediction (Angelopoulos & Bates, 2022)). Consider a set of independent and identically distributed (i.i.d.) calibration data, denoted as $\{(X_i, Y_i)\}_{i=1}^n$, along with a new test point $(X_{\text{test}}, Y_{\text{test}})$ sampled independently from the same distribution. Define a score function $s(x, y) \in \mathbb{R}$, where higher scores indicate poorer alignment between $x$ and $y$. Compute the calibration scores $s_1 = s(X_1, Y_1), \ldots, s_n = s(X_n, Y_n)$. For a user-defined confidence level $1 - \alpha$, let $\hat{q}$ represent the $\lceil (n + 1)(1 - \alpha) \rceil / n$ quantile of these scores. Construct the prediction set for the test input $X_{\text{test}}$ as:

$$\mathcal{C}(X_{\text{test}}) = \{y : s(X_{\text{test}}, y) \leq \hat{q}\}.$$

Assuming exchangeability, the prediction set $\mathcal{C}(X_{\text{test}})$ guarantees the marginal coverage property:

$$\mathbb{P}(Y_{\text{test}} \in \mathcal{C}(X_{\text{test}})) \geq 1 - \alpha.$$

Following the Lemma 1, we employ a conformal scoring function for safety verification, defined as:

$$s(X) = \hat{V}_{\hat{\pi}_\theta}(0, \hat{x}), \forall \hat{x} \in \mathcal{S}_{\tilde{\delta}},$$

where $\mathcal{S}_\delta$ denotes the set of states satisfying $\hat{V}_\theta(0, \hat{x}) \leq \delta$ and the score function measures the alignment between the induced safe policy and the auxiliary value function.

Next, we sample $N_s$ states from the safe set $\mathcal{S}_\delta$ and compute conformal scores for all sampled states. For a user-defined error rate $\alpha \in [0, 1]$, let $\hat{q}$ denote the $\frac{(N_s + 1)\alpha}{N_s}$th quantile of the conformal scores. According to (Vovk, 2012), the following property holds:

$$\mathbb{P}_{\hat{x} \in \mathcal{S}_{\tilde{\delta}}} \left( \hat{V}_{\hat{\pi}_\theta}(\hat{x}, 0) \leq \hat{q} \right) \sim \text{Beta}(N_s - l + 1, l), \tag{21}$$

where $l = \lfloor (N_s + 1)\alpha \rfloor$.

Define $E_s$ as:

$$E_s := \mathbb{P}_{\hat{x} \in \mathcal{S}_\delta} \left( \hat{V}_{\hat{\pi}_\theta}(\hat{x}, 0) \leq \hat{q} \right).$$

Here, $E_s$ is a Beta-distributed random variable. Using properties of cumulative distribution functions (CDF), we assert that $E_s \geq 1 - \epsilon_s$ with confidence $1 - \beta_s$ if the following condition is satisfied:

$$I_{1-\epsilon_s}(N - l + 1, l) \leq \beta_s, \tag{22}$$

where $I_x(a, b)$ is the regularized incomplete Beta function and also serves as the CDF of the Beta distribution. It is defined as:

$$I_x(a, b) = \frac{1}{B(a, b)} \int_0^x t^{a-1}(1 - t)^{b-1} \, dt,$$

where $B(a, b)$ is the Beta function. From (Olver et al., 2023)(8.17.5), it can be shown that $I_x(n-k, k+1) = \sum_{i=1}^{k} \binom{n}{i} x^i (1-x)^{n-i}$.

Then (22) can be rewritten as:

$$\sum_{i=1}^{l-1} \binom{N_s}{i} \epsilon_s^i (1-\epsilon)^{N_s-i} \leq \beta_s, \tag{23}$$

Thus, if Equation (23) holds, we can say with probability $1 - \beta_s$ that:

$$\mathbb{P}_{\hat{x} \in \mathcal{S}_{\tilde{\delta}}} \left( \hat{V}_{\hat{\pi}_\theta}(\hat{x}_i, 0) \leq \hat{q} \right) \geq 1 - \epsilon_s. \tag{24}$$

Now, let $k$ denote the number of allowable safety violations. Thus, the safety error rate is given by $\alpha_\delta = \frac{k+1}{N_s+1}$. Let $\hat{q}$ represent the $\frac{(N_s+1)\alpha_\delta}{N_s}$th quantile of the conformal scores. Since $k$ denotes the number of samples for which the conformal score is positive, the $\frac{(N_s+1)\alpha_\delta}{N_s}$th quantile of scores corresponds to the maximum *negative score* amongst the sampled states. This implies that $\hat{q} \leq 0$. From this and Equation (24), we can conclude with probability $1 - \beta_s$ that:

$$\mathbb{P}_{\hat{x} \in \mathcal{S}_\delta} \left( \hat{V}_{\hat{\pi}_\theta}(0, \hat{x}_i) \leq 0 \right) \geq 1 - \epsilon_s.$$

From Equation (4), it can be inferred that $\forall (t, \hat{x})$, $\hat{V}(0, \hat{x}_i) \leq \hat{V}_{\hat{\pi}_\theta}(\hat{x}_i, 0)$. Hence, with probability $1 - \beta_s$, the following holds:

$$\mathbb{P}_{\hat{x} \in \mathcal{S}_\delta} \left( \hat{V}(0, \hat{x}_i) \leq 0 \right) \geq 1 - \epsilon_s.$$

$\square$

## A.2. Theorem (3.2)

> **Theorem 3.2** (Performance Quantification Using Conformal Prediction) Suppose $\mathcal{S}^*$ denotes the safe states satisfying $V_\theta(0, x) < \infty$ (or equivalently $\hat{V}_\theta(0, \hat{x}^*) < \delta$) and $(0, x_i)_{i=1,...,N_p}$ are $N_p$ i.i.d. samples from $\mathcal{S}^*$. For a user-specified level $\alpha_p$, let $\psi$ be the $\frac{\lceil (N_p+1)(1-\alpha_p) \rceil}{N_p}$th quantile of the scores $(p_i := \frac{|V_\theta(0, x_i) - V_{\pi_\theta}(0, x_i)|}{C_{max}})_{i=1,...,N_p}$ on the $N_p$ state samples. Select a violation parameter $\epsilon_p \in (0, 1)$ and a confidence parameter $\beta_p \in (0, 1)$ such that:
>
> $$\sum_{i=0}^{l-1} \binom{N_p}{i} \epsilon_p^i (1-\epsilon_p)^{N_p-i} \leq \beta_p$$
>
> where, $l = \lfloor (N_p + 1)\alpha_p \rfloor$. Then, the following holds, with probability $1 - \beta_p$:
>
> $$\mathbb{P}_{x \in \mathcal{S}^*} \left( \frac{|V_\theta(0, x_i) - V_{\pi_\theta}(0, x_i)|}{C_{max}} \leq \psi \right) \geq 1 - \epsilon_p.$$
>
> where $C_{max}$ is a normalizing factor and denotes the maximum possible cost that could be incurred for any $x \in \mathcal{S}^*$.

*Proof.* To quantify the performance loss, we employ a conformal scoring function defined as:

$$p(x) := \frac{|V_\theta(0, x_i) - V_{\pi_\theta}(0, x_i)|}{C_{max}}, \forall x \in \mathcal{S}^*$$

where the score function measures the alignment between the induced optimal policy and the value function.

Next, we sample $N_p$ states from the state space $\mathcal{S}^*$ and compute conformal scores for all sampled states. For a user-defined error rate $\alpha_p \in [0, 1]$, let $\psi$ denote the $\frac{(N_p+1)\alpha_p}{N_p}$ quantile of the conformal scores. According to (Vovk, 2012), the following property holds:

$$\mathbb{P}_{x \in \mathcal{S}^*} \left( \frac{|V_\theta(0, x_i) - V_{\pi_\theta}(0, x_i)|}{C_{max}} \leq \psi \right) \sim \text{Beta}(N_p - l + 1, l),$$

where $l = \lfloor (N_p + 1)\alpha_p \rfloor$.

Define $E_p$ as:

$$E_p := \mathbb{P}_{x \in \mathcal{S}^*} \left( \frac{|V_\theta(0, x_i) - V_{\pi_\theta}(0, x_i)|}{C_{max}} \leq \psi \right).$$

Here, $E_p$ is a Beta-distributed random variable. Using properties of CDF, we assert that $E_p \geq 1 - \epsilon_p$ with confidence $1 - \beta_p$ if the following condition is satisfied:

$$I_{1-\epsilon_p}(N_p - l + 1, l) \leq \beta_p, \tag{25}$$

where $I_x(a, b)$ is the regularized incomplete Beta function. From (Olver et al., 2023)(8.17.5), it can be shown that $I_x(n - k, k + 1) = \sum_{i=1}^{k} \binom{n}{i} x^i (1 - x)^{n-i}$. Hence, Equation (25) can be equivalently stated as:

$$\sum_{i=1}^{l-1} \binom{N_p}{i} \epsilon_p^i (1 - \epsilon_p)^{N_p - i} \leq \beta_p \tag{26}$$

Thus, if Equation (26) holds, we can conclude with probability $1 - \beta_p$ that:

$$\mathbb{P}_{x \in \mathcal{S}^*} \left( \frac{|V_\theta(0, x_i) - V_{\pi_\theta}(0, x_i)|}{C_{max}} \leq \psi \right) \geq 1 - \epsilon_p.$$

$\square$

## A.3. Relationship between $\alpha$, $\beta$, and $\epsilon$

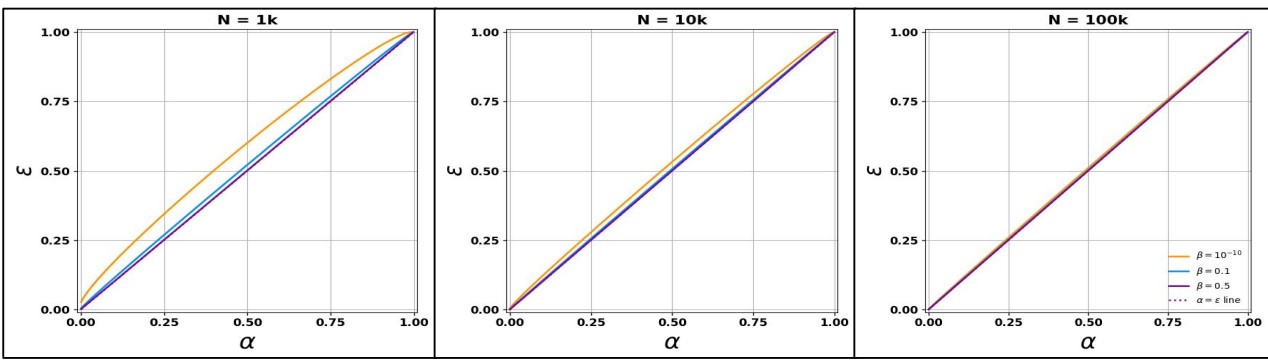

*Figure 7.* This figure shows the $\alpha$-$\epsilon$ plots for different numbers of verification samples, $N$, and different values of $\beta$.

The work (Vovk, 2012) states that a smaller number of samples leads to greater fluctuations in the conformal prediction calibration, meaning that if we redraw $N$ samples and repeat the conformal prediction process, we might get a different calibration result. This variance decreases as $N$ increases.Similarly, in our work, a small $N$ means that the value correction term $\delta$ might fluctuate each time the verification algorithm is executed. Therefore, to ensure a stable estimate of $\delta$, it is desirable to select a sufficiently large value of $N$.

Figure 7 presents the $\alpha - \epsilon$ plots for varying numbers of verification samples $N$ and different values of $\beta$. From the figure, we observe that as $N$ increases, the effect of $\beta$ diminishes, and the curve approaches the $\alpha = \epsilon$ line. Ideally, the user-specified safety error rate ($\alpha$) should closely match the safety violation parameter ($\epsilon$) while maintaining high confidence ($1 - \beta$ close to 1). Thus, selecting a larger $N$ enables a smaller $\beta$ while ensuring the alignment of $\alpha$ and $\epsilon$. Conversely, if $N$ is small, one must either compromise on the confidence parameter $\beta$ or accept that $\alpha$ will be lower than $\epsilon$, resulting in a more conservative upper bound on the safety rate.

## B. Additional Details the systems in the experiments

In this section, we will provide more details about the systems we have used in the experiments section 4.

## B.1. Efficient and Safe Boat Navigation

The states, $x$ of the 2D Boat system are $x = [x_1, x_2]^T$, where, $x_1, x_2$ are the $x$ and $y$ coordinates of the boat respectively. We define the step cost at each step, $l(t, x)$, as the distance from the goal, given by:

$$l(t, x) := \|x - (1.5, 0)^T\|$$

The cost function $C(t, x(t))$ is defined as:

$$C(t, x(t), \mathbf{u}) = \int_t^T l(t, x(t)) \, dt \; + \phi(x(T)) \tag{27}$$

where $T$ is the time horizon ($2s$ in our experiment), $l(t, x(t)) = \|x(t) - (1.5, 0)^T\|$ represents the running cost, and $\phi(x(T)) = \|x(T) - (1.5, 0)^T\|$ is the terminal cost. Minimizing this cost drives the boat toward the island.

Consequently, the (augmented) dynamics of the 2D Boat system are:

$$\dot{x_1} = u_1 + 2 - 0.5x_2^2$$
$$\dot{x_2} = u_2$$
$$\dot{z} = -l(t, x)$$

where $u_1, u_2$ represents the velocity control in $x_1$ and $x_2$ directions respectively, with $u_1^2 + u_2^2 \leq 1$ and $2 - 0.5x_2^2$ specifies the current drift along the $x_1$-axis.

The safety constraints are formulated as:

$$g(x) := max(0.4 - \|x - (-0.5, 0.5)^T\|, 0.5 - \|x - (-1.0, -1.2)^T\|) \tag{28}$$

where $g(x) > 0$ indicates that the boat is inside a boulder, thereby ensuring that the super-level set of $g(x)$ defines the failure region.

### B.1.1. GROUND TRUTH COMPARISON

We compute the Ground Truth value function using the Level-Set Toolbox (Mitchell, 2004) and use it as a benchmark in our comparative analysis. To facilitate demonstration, unsafe states are assigned a high value of 20 instead of $\infty$. The value function in this problem ranges from 0 to 14.76.

As illustrated in Figure 8, the value function obtained using our method closely approximates the ground truth value function. Notably, the unsafe region (highlighted in yellow) remains identical in both cases, confirming the safety of the learned value function. Furthermore, the mean squared error (MSE) between the two value functions is 0.36, which is relatively low given the broad range of possible values.

It is also worth mentioning that computing a high-fidelity ground truth value function on a $210 \times 210 \times 210$ grid using the Level Set Toolbox requires approximately 390 minutes. In contrast, our proposed approach learns the value function in 122 minutes, achieving a substantial speedup. This demonstrates that even for systems with a relatively low-dimensional state space, our method efficiently recovers an accurate value function significantly faster than grid-based solvers.

### B.1.2. EMPIRICAL VALIDATION OF THE $\alpha$-$\beta$-$\epsilon$ RELATIONSHIP AND CALCULATION OF SAFETY LEVELS

We conducted an experiment to empirically validate the theoretical relationship between $\alpha$, $\beta$, and $\epsilon$. The results are presented in Figure 9. The figure visualizes the relationship between theoretical and empirical safety metrics across varying levels of $\delta$, and includes the following elements:

- *Safety error rate* ($\alpha$, purple line) as a function of different $\delta$ levels. Computed on the calibration dataset as $\alpha = \frac{k+1}{N_s+1}$, where $k$ is the number of allowable safety violations and $N_s$ is the number of calibration samples.

- *Theoretical safety violation probability* ($\epsilon$, orange line) as a function of $\delta$. Derived using the theoretical relation in Equation (14).

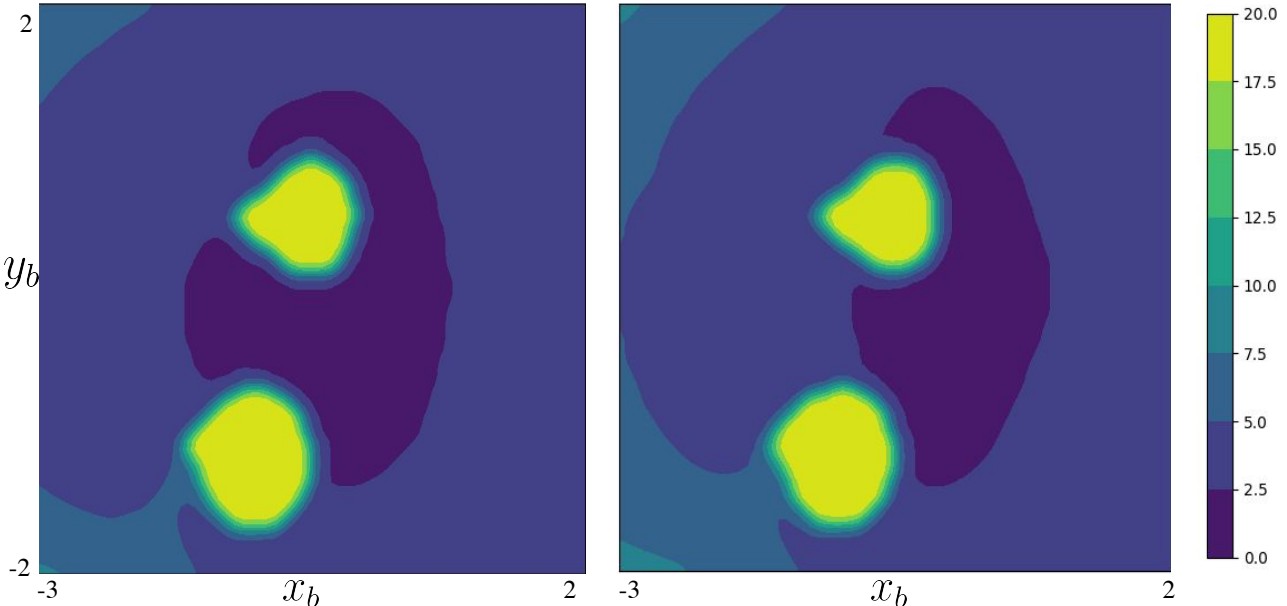

*Figure 8.* Heatmap of the value function for the ground truth (left) and our method (right). The yellow region represents the unsafe area. Our method successfully captures most of the safe set, indicating that it is not overly conservative while completely recovering the unsafe regions.

- *Empirical safety violation probability* (green points) as a function of $\delta$. Computed by sampling 3M initial states from the $\delta$-sublevel set of the learned value function, simulating rollouts, and measuring the observed safety violation rate. This serves as a practical estimate of system safety.

For this experiment, we set $N = 300k$ and $\beta = 10^{-10}$. As shown in the figure, the empirical violation rate remains consistently below the theoretical bound ($\epsilon$) across all values of $\delta$. This demonstrates that our method provides conservative and valid safety guarantees, confirming the soundness of the theoretical relationship in practice.

Additionally, from the $\delta$ vs $\epsilon$ plot, we can observe that the $\delta$ level approaches 0 as the $\epsilon$ values approach the chosen safety level of 0.001. Hence, we say that the sub-level set of the auxiliary value function, $\hat{V}(t, \hat{x})$ is safe with a probability of $1 - 0.001 = 0.999$.

### B.2. Pursuer vehicle tracking an evader

The state, $x$ of a ground vehicle (pursuer) tracking a moving evader is $x = [x_p, y_p, v, \Theta, x_e, y_e, v_{xe}, v_{ye}]^T$, where, $x_e, y_e, v, \Theta$ are position, linear velocity and orientation of the pursuer respectively, $x_e, y_e, v_{xe}, v_{ye}$ are the position and the linear velocities of the evader respectively. We define the step cost at each step, $l(t, x)$, as the distance from the goal, given by:

$$l(t, x) := \|(x_p(t), y_p(t))^T - (x_e(t), y_e(t))^T\|$$

and the terminal cost is $\phi(x(T)) = \|(x_p(T), y_p(T))^T - (x_e(T), y_e(T))^T\|$. The cost function $C(t, x(t))$ is defined as:

$$C(t, x(t), \mathbf{u}) = \int_t^T l(t, x(t)) \, dt + \phi(x(T)) \tag{29}$$

where $T$ is the time horizon ($1s$ in this experiment). Minimizing this cost aims to drive the pursuer toward the evader. Consequently, the (augmented) dynamics of the system is as follows:

$$\dot{x}_p = v \cos(\Theta), \quad \dot{y}_p = v \sin(\Theta), \quad \dot{v} = u_1, \quad \dot{\Theta} = u_2,$$
$$\dot{x}_e = v_{xe}, \quad \dot{y}_e = v_{ye}, \quad \dot{v}_{xe} = 0, \quad \dot{v}_{ye} = 0, \quad \dot{z} = -l(t, x),$$

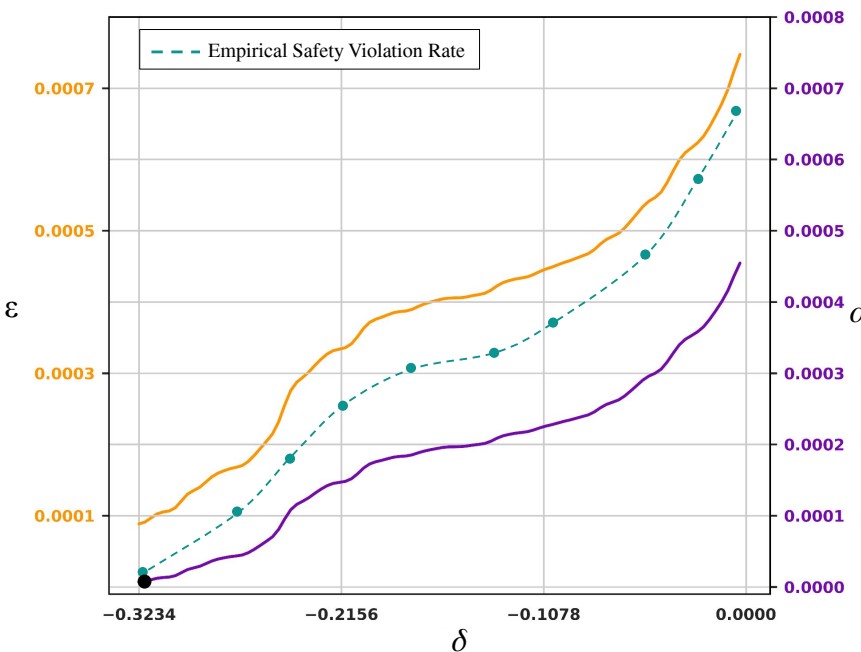

*Figure 9.* This figure presents a comparative analysis of the relationships between $\epsilon$–$\delta$, $\alpha$–$\delta$, and empirical safety violation rate–$\delta$. As observed, the empirical safety violation consistently remains below the theoretical bound, thereby supporting our theoretical guarantees. Furthermore, as $\epsilon$ decreases, the corresponding $\delta$ approaches zero, indicating that the learned value function incurs negligible safety violations.

where $u_1$ represents the linear acceleration control and $u_2$ represents angular velocity control.

The safety constraints are defined as:

$$g(x) := max(0.2 - \|x - (0.5, 0.5)^T\|, 0.2 - \|x - (-0.5, 0.5)^T\|, 0.2 - \|x - (-0.5, -0.5)^T\|,$$
$$0.2 - \|x - (0.5, -0.5)^T\|, 0.2 - \|x - (0.0, 0.0)^T\|, ))$$

which represents 5 obstacles of radius 0.2 units each.

### B.3. Multi-Agent Navigation

A multi-agent setting with 5 agents. The state of each agent $i$ is represented by $x_i = [x_{a_i}, y_{a_i}, x_{g_i}, y_{g_i}]$, tries to reach its goal while avoiding collisions with others. $(x_{a_i}, y_{a_i})$ denote the position of the $i$th agent, while $(x_{g_i}, y_{g_i})$ represent the goal locations for that agent. We define the step cost at each step, $l(t, x(t))$, as the mean distance of each agent from its respective goal, given by:

$$l(t, x(t)) := \frac{\sum_{i=1}^{5} \|(x_{a_i}(t), y_{a_i}(t)^T - (x_{g_i}(t), y_{g_i}(t))^T\|}{5}$$

The cost function $C(t, x(t), \mathbf{u})$ is defined as:

$$C(t, x(t), \mathbf{u}) := \int_{t}^{T} l(t, x(t)) \, dt + \phi(x(T)) \tag{30}$$

where $T$ is the time horizon ($2s$ in this experiment). Minimizing this cost aims to drive each agent towards its goal. Consequently, the (augmented) dynamics of the system is as follows:

$$\dot{x}_{a_i} = u_{1i}, \forall i \in \{1, 2, 3, 4, 5\}$$
$$\dot{y}_{a_i} = u_{2i}, \forall i \in \{1, 2, 3, 4, 5\}$$
$$\dot{x}_{g_i} = 0, \forall i \in \{1, 2, 3, 4, 5\}$$
$$\dot{y}_{g_i} = 0, \forall i \in \{1, 2, 3, 4, 5\}$$
$$\dot{z} = -l(t, x)$$

where $u_{1i}, u_{2i}$ represents the linear velocity control of each agent $i$. The safety constraints are defined as:

$$g(x(t)) := \max_{i,j=\{1,..,5\}, i \neq j} (R - \|(x_{a_i}, y_{a_i})^T - (x_{a_j}, y_{a_j})^T\|) \tag{31}$$

### B.3.1. COMPARISON OF MULTI-AGENT NAVIGATION WITH BASELINES

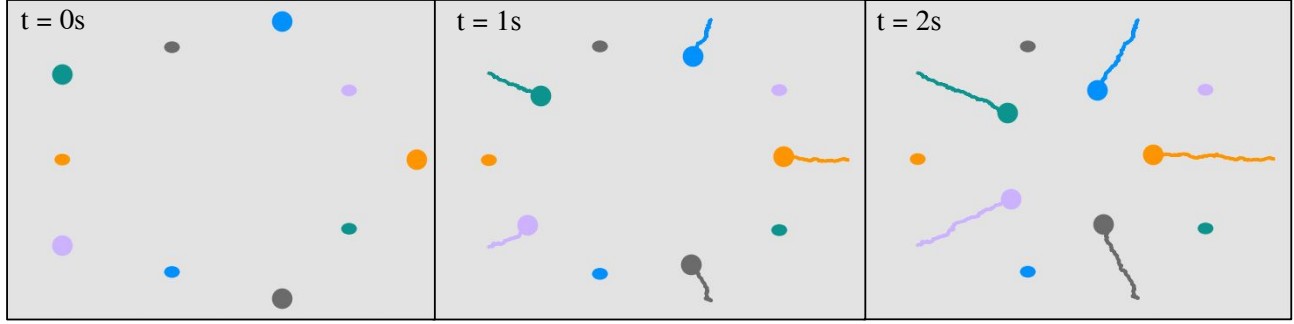

*Figure 10.* Snapshots of multi-agent navigation trajectories at different time instances using **MPPI**. The trajectories indicate that the agents adopt a **highly conservative strategy** to prevent collisions. Consequently, this leads to a **reduction in performance**, as the agents **end up very far from their respective goals**.

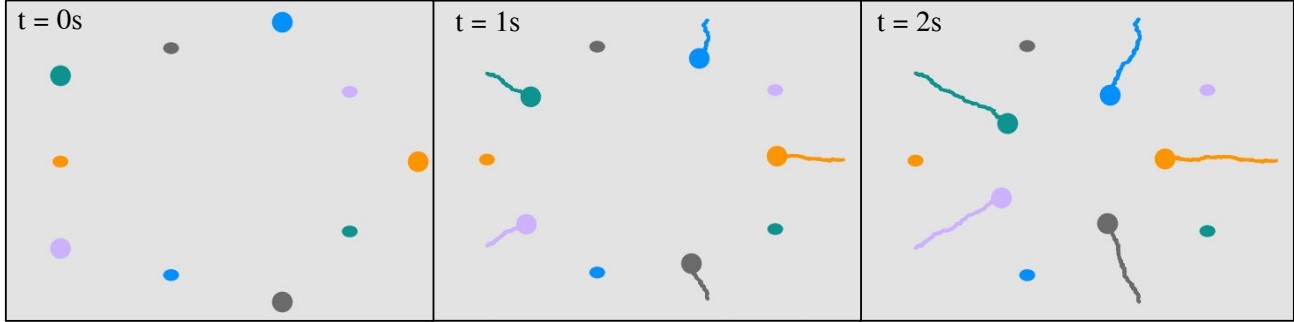

*Figure 11.* Snapshots of multi-agent navigation trajectories at different time instances using **MPPI-NCBF**. The observed trajectories demonstrate **suboptimal behavior similar to that of the MPPI policy**. Consequently, this results in high-performance costs, indicating its **inability to effectively co-optimize safety and performance.**

Figures 10, 11, and 12 illustrate the trajectories obtained by the baseline methods for the Multi-Agent Navigation problem. It can be observed that the trajectories obtained by MPPI and MPPI-SF are highly conservative, implying that these methods prioritize safety to mitigate potential conflicts among agents. In contrast, the policy derived from SAC-Lag fails to maintain safety, resulting in agent collisions. This indicates that as system complexity increases, the baseline methods tend to prioritize either safety or performance, leading to suboptimal behavior and safety violations. Conversely, the proposed approach effectively co-optimizes safety and performance, even in complex high-dimensional settings, achieving superior performance while ensuring safety. The visualization of the trajectories can be found on the project website[1].

---

[1] https://tayalmanan28.github.io/piml-soc/

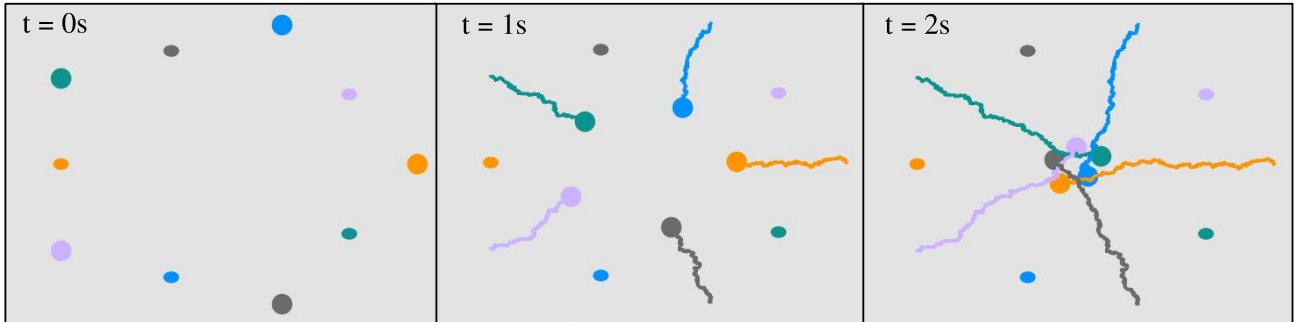

*Figure 12.* Snapshots of multi-agent navigation trajectories at different time instances using SAC-Lag. The trajectories indicate that agents **demonstrate less conservative behavior compared to MPPI and MPPI-NCBF, but they lead to collisions**. These **safety violations are critical** and cannot be disregarded, further **highlighting the limitations of the baseline methods in simultaneously optimizing safety and performance.**

## C. Implementation Details of the Algorithms

This section provides an in-depth overview of our algorithm and baseline implementations, including hyperparameter configurations and the cost/reward functions used in the baselines across all experiments.

### C.1. Experimentation Hardware

All experiments were conducted on a system equipped with an 11th Gen Intel Core i9-11900K @ 3.50GHz × 16 CPU, 128GB RAM, and an NVIDIA GeForce RTX 4090 GPU for training.

### C.2. Hyperparameters for the Proposed Algorithm

We maintained training settings across all experiments, as detailed below:

| Hyperparameter | Value |
|---|---|
| Network Architecture | Multi-Layer Perceptron (MLP) |
| Number of Hidden Layers | 3 |
| Activation Function | Sine function |
| Hidden Layer Size | 256 neurons per layer |
| Optimizer | Adam optimizer |
| Learning Rate | $2 \times 10^{-5}$ |
| **Boat Navigation** | . |
| Number of Training Points | 65000 |
| Number of Pre Training Epochs | 50K |
| No. of Training Epochs | 200K |
| **Pursuer Vehicle Tracking Evader** | . |
| Number of Training Points | 65000 |
| Number of Pre Training Epochs | 60K |
| No. of Training Epochs | 300K |
| **Multi Agent Navigation** | . |
| Number of Training Points | 65000 |
| Number of Pre Training Epochs | 60K |
| No. of Training Epochs | 400K |

*Table 1.* Hyperparameters for the proposed algorithm

## C.3. MPPI based baselines

For all the experiments we consider the MPPI cost term as follows:

$$C_{MPPI} = C(t, x(t), \mathbf{u}) + \lambda \max(g(x), 0) \tag{32}$$

where, $\lambda$ is the trade-off parameter, $C(t, x(t), \mathbf{u})$, $g(x)$ are the cost functions and safety functions as defined in Appendix B. Following is the list of hyperparameters we have used for MPPI experiments in all the cases:

| Hyperparameter | Value |
|---|---|
| Trade-off parameter ($\lambda$) | 100 |
| Planning Horizon | 20 |
| Softmax Lambda | 200 |
| No. of Rollouts | 8000 |

*Table 2.* Hyperparameters for MPPI Baselines

## C.4. SAC-Lag hyperparameters

For all the experiments, we consider the reward term as follows:

$$R_{SAC-Lag} = -C(t, x(t), \mathbf{u}) - \mathbb{I}_{g(x)>0} \times (100) + \mathbb{I}_{l(t,x(t))<0.1} \times (100) \tag{33}$$

where, $C(t, x(t), \mathbf{u})$, $g(x)$ are the cost functions and safety functions as defined in Appendix B. Table 3 provides the list of hyperparameters we have used for SAC experiments in all the cases.

| Parameter | Value |
|---|---|
| Policy Architecture | Multi-Layer Perceptron (MLP) |
| learning rate | $3 \times 10^{-4}$ |
| buffer size | $1,000,000$ |
| learning starts | $10,000$ |
| batch size | $256$ |
| Target network update rate ($\tau$) | $0.005$ |
| Discount factor ($\gamma$) | $0.99$ |
| **Boat Navigation** | . |
| Number of Training Steps | 1,000,000 |
| **Pursuer Vehicle Tracking Evader** | . |
| Number of Training Steps | 2,500,000 |
| **Multi Agent Navigation** | . |
| Number of Training Steps | 1,000,000 |

*Table 3.* General Hyperparameters of SAC in our experiments

## C.5. PPO-Lag hyperparameters

For all the experiments, we consider the reward term as follows:

$$R_{PPO-Lag} = -C(t, x(t), \mathbf{u}) - \mathbb{I}_{g(x)>0} \times (100) + \mathbb{I}_{l(t,x(t))<0.1} \times (100) \tag{34}$$

where, $C(t, x(t), \mathbf{u})$, $g(x)$ are the cost functions and safety functions as defined in Appendix B. Table 4 provides the list of hyperparameters we have used for PPO experiments in all the cases.

| Parameter | Value |
|---|---|
| Policy Architecture | Multi-Layer Perceptron (MLP) |
| learning rate | $3 \times 10^{-4}$ |
| buffer size | $1,000,000$ |
| learning starts | $10,000$ |
| batch size | $256$ |
| Target network update rate ($\tau$) | $0.005$ |
| Discount factor ($\gamma$) | $0.99$ |
| **Boat Navigation** | . |
| Number of Training Steps | 1,000,000 |
| **Pursuer Vehicle Tracking Evader** | . |
| Number of Training Steps | 2,500,000 |
| **Multi Agent Navigation** | . |
| Number of Training Steps | 1,000,000 |

*Table 4.* General Hyperparameters of PPO in our experiments

## C.6. CPO hyperparameters

For all the experiments, we consider the reward term as follows:

$$R_{CPO} = -C(t, x(t), \mathbf{u}) + \mathbb{I}_{l(t,x(t))<0.1} \times (100) \tag{35}$$

where $C(t, x(t), \mathbf{u})$, $g(x)$ are the cost functions and safety functions as defined in Appendix B. For the CPO implementation, we have used the training settings used in (Chen et al., 2020). Table 5 provides the list of hyperparameters we have used for CPO experiments in all the cases.

| Parameter | Value |
|---|---|
| Policy Architecture | Multi-Layer Perceptron (MLP) |
| Batch Size | 128 |
| Target KL Divergence | 0.01 |
| Entropy Coefficient | 0.0 |
| Reward Discount Factor ($\gamma$) | 0.99 |
| Cost Discount Factor ($\gamma_c$) | 0.99 |
| GAE Lambda ($\lambda$) | 0.95 |
| Cost GAE Lambda ($\lambda_c$) | 0.95 |
| Critic Norm Coefficient | 0.001 |
| Penalty Coefficient | 0.0 |
| Conjugate Gradient Damping | 0.1 |
| Conjugate Gradient Iterations | 15 |
| Actor Hidden Sizes | $[256, 256]$ |
| Critic Hidden Sizes | $[256, 256]$ |
| Critic Learning Rate | 0.001 |
| **Boat Navigation** | . |
| Number of Training Steps | 10,000,000 |
| **Pursuer Vehicle Tracking Evader** | . |
| Number of Training Steps | 25,000,000 |
| **Multi Agent Navigation** | . |
| Number of Training Steps | 10,000,000 |

*Table 5.* CPO Hyperparameters from OmniSafe Configuration used for our experiments

