# OpenReview forum: "A Physics-Informed Machine Learning Framework for Safe and Optimal Control of Autonomous Systems"
_ICML.cc/2025/Conference — ICML 2025 poster_

### Official Review · Reviewer_BKAj · 2025-03-13

**Overall Recommendation:** 3

**Summary:**

This work develops a physics-informed learning approach for constrained optimal control problem, where the performance objectives are represented using a cost function and safety conditions are formulated using state constraints. A conformal prediction-based safety verification approach is developed with probabilistic error bound on performance degradation. Three simulation examples are provided to demonstrate the performance of the approach.

**Claims And Evidence:**

The contribution is clear and mostly supported by the mathematical exposition and simulation results. The paper makes an interesting contribution relevant to the machine learning, controls, and robotics communities. However, claiming that the paper develops a novel physics-informed machine learning framework is misleading because the paper does not provide any fundamentally new physics-informed machine learning method; the development is mostly a direct application of well-known physics-informed machine learning methods.

**Essential References Not Discussed:**

The following reference is relevant and needs to be discussed in the introduction:

Fotiadis, F. and Vamvoudakis, K.G., 2023, December. A Physics-Informed Neural Networks Framework to Solve the Infinite-Horizon Optimal Control Problem. In 2023 62nd IEEE Conference on Decision and Control (CDC) (pp. 6014-6019). IEEE.

**Experimental Designs Or Analyses:**

1. The experimental methodology is mostly appropriate. The results are shown to be applicable for high-dimensional systems (20 dimensional system in the multi-agent navigation case).
2. The paper assumes known dynamics which by itself is not problematic, but what would happen if the dynamics have modeling uncertainties and/or disturbances? In practice, this is usually the case, so the experiments need to account for such modeling uncertainties to be realistic.
3. One of the benefits of using physics-informed neural networks is that one can obtain good performance with fewer training samples. The experiments use 65,000 training points. What would happen if fewer datapoints are used.

**Methods And Evaluation Criteria:**

The methodology and evaluation criteria is appropriate.

**Other Comments Or Suggestions:**

None

**Other Strengths And Weaknesses:**

None

**Questions For Authors:**

None, please see the previous comments

**Relation To Broader Scientific Literature:**

The key contributions are related to approximate dynamic programming (ADP) and physics-informed machine learning research. Notably, value function approximation using neural networks is not new, the authors can discuss more relevant ADP papers/optimal control papers to put contribution in context. The authors can also discuss more physics-informed machine learning literature.

**Theoretical Claims:**

1. The theoretical claims (Theorems 1 and 2) appear correct.

2. However, it is not clear if the cost function and safety constraints need to be convex. Are the authors claiming the method applies to non-convex optimal control problems? Some theoretical exposition in this regard would improve the quality of the paper.

3. The training loss in (8) and (9) do not appear to have a regularizing term (e.g., Tikhonov regularization). In physics-inforrmed learning, having a regularizing term in the loss is important as it can prevent the value function estimate from learning a trivial solutions that would yield a zero PDE residual error and also mitigate overfitting. Furthermore, the authors could also include penalty terms to penalize undesirable values of \hat{V}_{theta}, use it as a constraint in the training, or encode hard constraints in the NN architecture itself (see Lyapunov-Net reference below).

4. The paper claims the auxiliary value function is a unique continuous viscosity solution to the HJB, citing the reference (Altarovici et al., 2013). However, (Altarovici et al., 2013) makes this claim under multiple assumption (see A1-A4 in that result), one of which is that the dynamics f(x,u) be globally Lipschitz in x which highly restrictive for most practical applications). The paper makes no mention of these assumptions. Furthermore, the viscosity solutions are only guaranteed to be continuous and there are no guarantees on its differentiability. However, the control policy is dependent on the derivative of the auxiliary value function. Without differentiability, the feedback law might not be well-defined.

5. The approach using safety verification with conformal prediction is interesting.

References:

Gaby, N., Zhang, F. and Ye, X., 2022, December. Lyapunov-net: A deep neural network architecture for lyapunov function approximation. In 2022 IEEE 61st Conference on Decision and Control (CDC) (pp. 2091-2096). IEEE.

Altarovici, A., Bokanowski, O. and Zidani, H., 2013. A general Hamilton-Jacobi framework for non-linear state-constrained control problems. ESAIM: Control, Optimisation and Calculus of Variations, 19(2), pp.337-357.

---

> ### Author Rebuttal · Authors · 2025-03-31
>
> # Method application to non-convex OCP
>
> Our method extends beyond convex OCPs. While a convex cost function leads to a convex epigraph formulation that standard optimizers can efficiently solve, our approach is not restricted to them.  By leveraging dynamic programming to solve the epigraph formulation, we obtain a general HJB-VI, which is well-suited for handling both convex and non-convex settings. In fact, our experiments explicitly demonstrate this capability by incorporating non-convex and even disconnected safety constraint sets.
>
> # Tackling Model Uncertainty/Disturbances
>
> This study pioneers a learning-based approach to jointly optimize safety and performance using HJ Reachability, assuming accurate model information. Extending it to account for model uncertainty is a key future direction. As an initial step, we can follow [3] to handle dynamics uncertainty, modelling it as $\dot{x}(t) = f(x(t), u(t), d(t))$, where $d(t)$ captures learning uncertainty, unmodeled dynamics, or disturbances. We then define a robust auxiliary value function $\hat{V}$:
>
> \begin{equation}
>     \hat{V}(t, x(t), z) = \min_{\textbf{u}} \max_{\textbf{d}} \max \{C(t, x(t), \textbf{u}, \textbf{d}) -z, \max_{s \in [t, T]}g(x(s)) \}.
> \end{equation}
> The auxiliary value function optimizes performance cost under worst-case uncertainty while guaranteeing safety. As with the non-robust formulation, it is characterized as the unique viscosity solution of the HJI-VI:
> \begin{equation}
> \min\Bigl(-\partial_t \hat{V} - \min_{u} \max_{d} \langle \nabla_{\hat{x}}\hat{V}(t, \hat{x}), \hat{f}(\hat{x}, u, d)\rangle ,\hat{V} - g(x)\Bigr) = 0,
> \end{equation}
> To incorporate uncertainty (e.g., model errors or adversarial settings), we solve the PDE using our proposed method. As an initial validation, we apply it to the Pursuer-Evader problem with uncertainty in the evader model, capturing potential variability in human behavior. Figure 1 (https://shorturl.at/sxkvw) illustrates this: dashed lines show undisturbed trajectories, while solid lines depict uncertain cases where the evader moves unpredictably. The pursuer adapts with sharper turns, ensuring robust tracking despite uncertainty. Future work will further explore uncertainty handling.
>
> # Training loss does not have a regularizing term
>
> To mitigate trivial solutions, we employ the adaptive loss balancing strategy from [5], which dynamically adjusts $\lambda$ at each training step to balance the boundary loss and PDE loss. This ensures both objectives are enforced jointly.
> As future work, we plan to incorporate explicit regularization, such as Tikhonov ($L_2$) regularization, to prevent overfitting and improve generalization. Additionally, we will explore hard constraint enforcement techniques within the neural network architecture—such as those proposed in [1,2]—to exactly satisfy boundary conditions and enhance the fidelity of the learned value function.
>
> # Missing Assumptions
>
> We appreciate the reviewer for highlighting the missing assumptions. In the revised manuscript, we will explicitly state those assumptions. In practice, assuming local Lipschitz continuity is often sufficient, particularly when the value function needs to be computed over a compact state space, as is often the case with PINN training (including our work). Under this assumption, the continuity and the well-posedness of the HJB solution can still be ensured without requiring global Lipschitz continuity. We will revise the manuscript accordingly to clarify the assumptions and better contextualize their relevance in practical settings.
>
> # Differentiability of $\hat{V}$
>
> We approximate the value function using a PINN with *sinusoidal* activations, ensuring smoothness and differentiability of the value function. Thus, the feedback control law, derived from the value function gradients, is well-defined and implementable.
>
> # Fewer Training Samples to train PINNs
>
> Since our approach is entirely self-supervised, generating training data is computationally inexpensive. Thus, we use a large number of training points (65K) to promote stability and convergence during training. However, we acknowledge that fewer points may be beneficial in high-dimensional cases to reduce memory and computation. While not explored here, we will investigate trade-offs between sample efficiency and performance in future work.
>
> # Essential References Not Discussed
>
> We thank the reviewer for the suggestion. We will include [4] as well as an analysis of other PIML approaches for HJ Reachability and optimal control in the revised manuscript.
>
> # References
>
> [1] Gaby et al. (2022). Lyapunov-net for Lyapunov function approximation. IEEE CDC.
>
> [2] Singh et al. (2025). Safety boundary conditions in neural reachable tubes. IEEE ICRA.
>
> [3] Bansal et al. (2017). Hamilton-Jacobi reachability. IEEE CDC.
>
> [4] Fotiadis & Vamvoudakis (2023). PINNs for infinite-horizon control. IEEE CDC.
>
> [5] Wang et al. (2021). Gradient flow pathologies in PINNs. SIAM J. Sci. Comput.

---

### Official Review · Reviewer_a3t3 · 2025-03-13

**Overall Recommendation:** 4

**Summary:**

In this work, the Authors propose a novel framework for the certified safety of autonomous agents based on the combination of the epigraph-based formulation of the optimal control problem, DeepReach, and conformal predictions. They test the proposed framework in three simulated environments and show the advantage of the proposed framework over the models that either mostly focus on performance or mostly focus on safety.

**Claims And Evidence:**

The work follows the standard routine in the field of the safety of autonomous agents. As a result, the claims here are supported by the standard means, previously validated in the related research.

**Essential References Not Discussed:**

I feel like some further, newer references could be added to the Introduction to better reflect the current state of the research in the field. Perhaps, review papers such as the one on The Safety Filter by Jaime Fisac’s group could be a good addition here.

**Experimental Designs Or Analyses:**

The testing scenarios and baselines here appeared overly simple to me. While there’s nothing wrong with this, typically the related research in the field has used stronger benchmarks, both simulated and data-driven, and considered stronger baselines. There have been lots of advancements in the field lately; it would be nice to compare some of those – per Authors’ choice – to the proposed model.

**Methods And Evaluation Criteria:**

I found this work to present a step in the right direction. Naturally, the next step after having DeepReach, a neural network model for solving the Hamilton-Jacobi Reachability problem, was to extend it with the safety guarantees accounting for the errors potentially made by the deep learning model. I have, however, the following concern regarding the type of the guarantees provided by the conformal predictions, as follows. Equation 15 formulates the probability of the safety violation per state $\hat{x}$ in $S_\delta$. It would seem that, for longer trajectories, this quantity would accumulate, making the target values of $\epsilon$ low. How low exactly and how strong the safety requirements would be in that case is a question that, I believe, is important to investigate and answer. At the same time, I would like to notice that this form of guarantee is a huge step forward compared to the cumulative guarantees used previously with conformal predictions, as they did not preclude the possibility of short-term safety violations.

Additionally, I would like to note that the very same machinery applied here to the cost estimate in Theorem 2 makes perfect sense to me. As point violations of this estimate are not safety-critical, conformal predictions here offer a nice way to make that estimation that, on average, should be correct.

**Other Comments Or Suggestions:**

N/A

**Other Strengths And Weaknesses:**

Strength: the text is mostly clearly written. Combined with a well-structured research project, it is a pleasure to read.

Weakness: the methods are not fully documented. It is important to detail, in the Methods section, the details of the training (beyond the structure of the neural network that is present in Appendix).

**Questions For Authors:**

- Why is the i.i.d. assumption made in both theorems?

- Why is the assumption of stability (i.e. that $\delta$ does not grow over time) made in the model?

- What would be the relevant values of $\epsilon$ and $\beta$ provided that the quantity in Equation 5 would accumulate over time and states traversed?

- What was the training procedure for the curriculum learning?

- Could you perform an experiment to show that the theoretically derived relation between $\alpha$, $\beta$, and $\epsilon$ holds?

I am happy to update my score based on the answers to these questions.

__________________________________

Post-rebuttal: my questions and concerns were mostly addressed; raising the score.

**Relation To Broader Scientific Literature:**

The work is rooted in the host of prior literature. Specifically, the Hamilton-Jacobi Reachability, the epigraph formulation of the optimal control problem, DeepReach, and conformal predictions are at this point all well-established tools used in the certified safety of autonomous systems. The combination of DeepReach and the conformal predictions that is aimed to reinstate the certifiable safety in the model that otherwise computed a solution through a deep neural network, is a novel and natural step forward for that line of research.

**Theoretical Claims:**

I have checked the theoretical claims. While the theory looks mostly correct and follows the standard practices in the field, my biggest question is related to the i.i.d. assumption in both Theorems 1 and 2. While I understand that this assumption is central for using the conformal predictions (e.g. through Lemma 1 of Supplementary Material), I am not sure if its use is justified here for the following reason. While under this assumption $N_s$ is sampled i.i.d. from $S_\delta$, this is arguable not where the states of interest reside. The states of interest, balancing safety and performance, seem likely to reside on the edge of the set $S_\delta$ ($=\partial S_\delta$); thus the i.i.d estimation may be biased.

If I am correct, two ways to mitigate that potential issue are:

- to set $\alpha=1$. If it wouldn’t affect the value of $\beta$ as a function of $\epsilon$ terribly, that would allow circumnavigating the entire machinery of the conformal predictions, instead converging to a trivial limit-case result from combinatorics (which was also a nice way for me to check that the equations are correct)!

- to perform an empirical study where the numbers of safety violations would be measured as a function of $\alpha$ and $\beta$ to see if the simulated experiment reproduces the theoretically predicted trend.

I suggest trying both options as they are easy to try and have the potential of making the results here stronger.

Another question that I have is: the framework seems to assume that the deviations of the neural network model solutions from the actual solutions are local and do not accumulate over time. It would be nice to hear the argument regarding why this might be the case.

---

> ### Author Rebuttal · Authors · 2025-03-31
>
> # Clarification on Curriculum Learning
> As discussed in Sec. 3.1, we first pre-train the DNN to learn the value function at the terminal time ($t=T$)—i.e., the boundary condition of the HJB-VI—using $\lambda = 0$. We then apply curriculum learning, gradually decreasing $t$ from $T$ to $0$, so the terminal value function propagates backward per the HJB-VI. At each iteration, we uniformly sample $N$ states and time points from [$t, T$]) and train using the loss in Eq. (10).
>
> # Clarification on Verification Procedure
> We would like to provide further clarification on the proposed verification procedure. First, we estimate the induced policy $\hat{\pi_{\theta}}$ from the learned value function $\hat{V_{\theta}}$. Next, we generate 300K rollouts from $t = 0$ to $t = T$, starting from various initial states $\hat{x_0}$, and compute the corresponding rollout costs $\hat{V}_{\hat{\pi}}$. We then apply conformal prediction to compute the correction $\delta$, which represents the safety correction over the **entire trajectory** from $t = 0$ to $t = T$. Thus, the violation probability $\epsilon_s$ is defined for the full trajectories starting from $\hat{x_0}$, not individual states.
>
> # Why $\delta$ Does Not Grow Over Time
> As discussed above, the correction term $\delta$ is based on the rollout cost evaluated over the entire trajectory from $t = 0$ to $t = T$. If, instead, the cost were evaluated starting later in the trajectory, i.e., from $t = t_0$ to $t = T$ for some $t_0 > 0$-the trajectory would typically experience fewer safety violations, since early violations are excluded from the cost computation. Thus, the magnitude of $\delta$ at the beginning of the trajectory is greater than or equal to its magnitude at any later time. This property ensures that the initial correction term is conservative enough to account for worst-case safety costs, leading to robust safety guarantees.
>
> # i.i.d. Assumption in Theorems
> We assume i.i.d. sampling over the safe set as we lack prior knowledge of the test-time initial state distribution. This ensures generality in our safety guarantees. If a specific initial state distribution is of interest, it can be used for sampling instead to obtain tighter safety estimates. To reduce reliance on distributional assumptions, one can also adopt a worst-case approach by setting $\alpha = 0$, as the reviewer suggested.
>
> # Potential bias from sampling near $\partial S_{\delta}$
> While states delicately balancing safety and performance often reside near $\partial S_{\delta}$, learning errors can occur anywhere in $S_{\delta}$. Sampling only near the boundary may underestimate violations in the interior, which can be problematic if the initial states of interest during the test time are inside $S_{\delta}$. In the absence of prior distribution knowledge, we adopt uniform sampling across $S_{\delta}$ to capture violations comprehensively.
>
> # Experiment Validating $\alpha$-$\beta$-$\epsilon$ Relationship
> To empirically validate this relationship, we conducted an experiment in the Boat Navigation case study (Sec. 4.1). Figure 1 (https://shorturl.at/VYYai) illustrates:
>
> - Safety error rate ($\alpha$) computed as $\alpha = \frac{k+1}{N_s+1}$ (Purple).
> - Theoretical safety violation probability ($\epsilon$) derived from Equation (14) (Orange).
> - Empirical safety violation probability, estimated via sampling 3 million initial states and simulating their rollouts (Green).
>
> We use $N = 300K$ and $\beta = 10^{-10}$. We note that empirical violation rates consistently remain below theoretical bounds, confirming the validity of our approach.
>
> # Setting $\alpha = 1$ and Circumventing Conformal Prediction
> We acknowledge that setting $\alpha = 0$ (zero violations) indeed eliminates the need for conformal prediction. To test this, we mark the black point in Figure 1 above that corresponds to $\alpha = 0$ and $\delta$ level of $-0.3234$, yielding nearly 100\% safety. As expected, this approach reduces the volume of the resulting safe set, but we still obtain a sizable safe set that accounts for worst-case violations, as suggested by the reviewer. In contrast, $\delta = 0$ achieves ~99.9\% safety, reducing conservatism but at a slight safety compromise. In this sense, our framework can be viewed as a robust generalization of this extreme-case approach, offering users the flexibility to choose their preferred level of conservatism based on the application's safety-performance requirements.
>
> # Relevant $\epsilon$ and $\beta$, if quantity in eq(15) Accumulate Over Time
> Since all the analysis is performed for the entire trajectory starting from a given state, $\epsilon$ and $\beta$ already account for time accumulation.
>
> # Additional Baselines
> We kindly refer the reviewer to the discussion on additional baselines with reviewer: Rqif.
>
> # Essential References Not Discussed
> We appreciate the reviewer’s suggestion and will include [1] in the revised manuscript.
> [1] Hsu et al. The safety filter: A unified view...

---

> > ### Comment · Reviewer_a3t3 · 2025-04-03
> >
> > I was really excited to read your rebuttal and to see the way my comments were thoroughly addressed.
> >
> > While I'm still not sure about the i.i.d assumption and would encourage the Authors to further think about it (that is, balancing safety and efficiency may put the trajectories close to the boundaries of the safe set), the empirical evidence provided by the Authors confirms the theory, at least, on the considered task.
> >
> > With that, I am raising my score. Thanks for all the good work!

---

> > > ### Author Response · Authors · 2025-04-06
> > >
> > > Thank you for your thoughtful feedback and for increasing your score! We appreciate your detailed comments and are glad that our clarifications addressed your concerns. Your constructive engagement has been invaluable, and we look forward to further refining our work based on your insights.

---

### Official Review · Reviewer_7cLD · 2025-03-14

**Overall Recommendation:** 3

**Summary:**

This paper proposes a novel Physics-Informed framework to address the co-optimization problem of safety objectives and performance objectives for Constrained Reinforcement Learning (CRL). The paper reformulates the co-optimization problem as a state-constrained optimal control problem (SC-OCP) with epigraph formulation. Moreover, the authors propose an algorithmic approach to learning the SC-OCP value function from which the policy is induced. At last, this paper proposes a conformal prediction-based verification strategy to provide a probabilistic guarantee of safety and performance degradation. The proposed framework is demonstrated on several nonlinear control tasks.

**Claims And Evidence:**

Yes

**Essential References Not Discussed:**

No

**Experimental Designs Or Analyses:**

Yes.

**Methods And Evaluation Criteria:**

Yes

**Other Comments Or Suggestions:**

Please see the questions.

**Other Strengths And Weaknesses:**

Strengths:
1. The targeted problem is well-motivated, and the solution is novel.
2. The reformulation of the CRL problem is insightful, and the derived physics-informed loss function for the learning value function is mathematically sound.
3. The theoretical analysis and empirical study support the claimed contributions.

Weaknesses:
1. Some technical details need more clarification. Please see the questions

**Questions For Authors:**

1. How is the policy $\pi_\theta$ synthesized? The searched policy should minimize equation 11; however, how the policy is learned is unclear. The authors should provide more details, such as what algorithm is used and how policy is parameterized.
2. How is the parameter $\lambda$ determined in equation 10? How does $\lambda$ affect the safety and performance? It would be nice if the authors could provide more insights.
3. To achieve a high probabilistic guarantee, the proposed verification methods need to densely sample the state space (300K), which is non-trivial. Can you provide more discussion on the practicality and potential limitations of real-world applications?

**Relation To Broader Scientific Literature:**

The proposed contributions are related to Constrained  Reinforcement Learning.

**Theoretical Claims:**

Partially, I checked all equations and theoretical statements except for the proofs for Theorem 3.2

---

> ### Author Rebuttal · Authors · 2025-03-31
>
> # How is the policy $\pi_{\theta}$ synthesized?
>
> The final policy $\pi_{\theta}(t, x)$ is synthesized by first determining the optimal $z^*$ by solving the following optimization problem for any $(t,x)$:
>
> \begin{equation}
>     \begin{aligned}
>     z^* = &\arg\min_{z \in \mathbb{R^+}}  \; z  \\
>     \text{s.t.} & \; \hat{V}_{\theta}(t,x,z) \leq \delta,
> \end{aligned}
> \end{equation}
>
> For efficiency purposes, we solve the above optimization problem via binary search on $z$.
> Once $z^*$ is determined, the optimal control policy is the policy that minimizes the Hamiltonian ($H(t,\hat{x})$) [1] at the corresponding augmented state $(x, z^*)$, denoted as $\hat{x}^*$:
>
> \begin{equation}
>     H(t,\hat{x}^*) = \min_{u\in \mathcal{U}} \langle\nabla \hat{V}_{\theta}(t, \hat{x}^*), \hat{f}(\hat{x}^*, u)\rangle.
> \end{equation}
>
> This results in the final policy $\pi_{\theta}$, as provided in Equation (17) of the submitted manuscript:
>
> \begin{equation}
>     \pi_{\theta}(t,x) = \arg \min_{u\in \mathcal{U}} \langle\nabla \hat{V}_{\theta}(t, \hat{x}^*), \hat{f}(\hat{x}^*, u)\rangle.
> \end{equation}
>
> Intuitively, the policy $\pi_{\theta}$ at any state $x$ follows the minimum cost path, which is given by the gradient descent direction of the auxiliary value function.
> The overall policy inference process takes $2ms$ on our standard GPU system.
>
> # How is the parameter $\lambda$ determined in equation 10? How does it affect the safety and performance?
>
> For a given partial differential equation (PDE), multiple solutions may exist, and uniqueness is ensured only when an appropriate boundary condition is imposed. The parameter $\lambda$ regulates the enforcement of this boundary condition. If $\lambda$ is too small, the boundary function may not be adequately learned, resulting in an inaccurate solution. To address the sensitivity of the learned solution to $\lambda$, we employ the adaptive loss restructuring approach from [2], which dynamically updates $\lambda$ at each iteration. The updates are directly proportional to the ratio of the boundary loss gradient to the PDE loss gradient. Intuitively, a small ratio results in a lower $\lambda$, prioritizing the PDE loss, whereas a larger ratio increases $\lambda$, emphasizing boundary condition enforcement.
>
>
> # To achieve a high probabilistic guarantee, the proposed verification methods need to densely sample the state space (300K), which is non-trivial. Can you provide more discussion on the practicality and potential limitations of real-world applications?
>
> The work [3] by Vovk states that a smaller number of samples leads to greater fluctuations in the conformal prediction calibration, meaning that if we redraw $N$ samples and repeat the conformal prediction process, we might get a different calibration result.
> This variance decreases as $N$ increases.
> Similarly, in our work, a small $N$ means that the value correction term $\delta$ might fluctuate each time the verification algorithm is executed. Therefore, to ensure a stable estimate of $\delta$, it is desirable to select a sufficiently large value of $N$.
>
> Additionally, Figure 1 in https://shorturl.at/8vgih presents the $\alpha-\epsilon$ plots for varying numbers of verification samples $N$ and different values of $\beta$. From the figure, we observe that as $N$ increases, the effect of $\beta$ diminishes, and the curve approaches the $\alpha = \epsilon$ line. Ideally, the user-specified safety error rate ($\alpha$) should closely match the safety violation parameter ($\epsilon$) while maintaining high confidence ($1-\beta$ close to 1).
> Thus, selecting a larger $N$ enables a smaller $\beta$ while ensuring the alignment of $\alpha$ and $\epsilon$. Conversely, if $N$ is small, one must either compromise on the confidence parameter $\beta$ or accept that $\alpha$ will be lower than $\epsilon$, resulting in a more conservative upper bound on the safety rate.
>
>
> **References**
>
> [1] S. Bansal, M. Chen, S. Herbert, and C. J. Tomlin, "Hamilton-Jacobi reachability: A brief overview and recent advances," 2017 IEEE 56th Annual Conference on Decision and Control (CDC), Melbourne, VIC, Australia, 2017, pp. 2242-2253, doi: 10.1109/CDC.2017.8263977.
>
> [2] S. Wang, Y. Teng, and P. Perdikaris, “Understanding and mitigating gradient flow pathologies in physics-informed neural networks,” SIAM Journal on Scientific Computing, vol. 43, no. 5, pp. A3055–A3081, 2021.
>
> [3] V. Vovk, “Conditional validity of inductive conformal predictors,”
> 2012. [Online]. Available: \\ https://arxiv.org/abs/1209.2673

---

### Official Review · Reviewer_Rqif · 2025-03-15

**Overall Recommendation:** 3

**Summary:**

The paper addresses the challenge of simultaneously optimizing performance and safety in autonomous systems by formulating it as a state-constrained optimal control problem. The key contribution is a physics-informed machine learning (PIML) framework that efficiently approximates the Hamilton-Jacobi-Bellman (HJB) equation, capturing the optimal value function under hard safety constraints. By embedding known system dynamics into the learning process, the approach mitigates the computational challenges of solving HJB equations in high-dimensional settings. Additionally, the authors introduce a conformal prediction-based verification method to quantify learning errors and provide probabilistic safety guarantees. The learned value function is statistically corrected to ensure safety with a high-confidence bound, and a separate conformal prediction step quantifies potential performance degradation. Empirical evaluations on three autonomous control tasks demonstrate that the framework learns controllers that maintain safety constraints while achieving near-optimal performance.

## update after rebuttal
I thank the authors for their response. My questions have been largely answered. New baselines and experimental results for scalability are appreciated and will make a good addition to the paper. For the new safe RL baselines, please include experimental results for all tasks in the paper. I updated my overall recommendation accordingly.

**Claims And Evidence:**

Most claims in the paper are reasonable and supported by empirical results.

The authors claim that their framework scales to complex, high-dimensional systems where classical Hamilton-Jacobi (HJ) or grid-based methods become computationally infeasible. While their experiments include tasks with moderately high-dimensional state spaces, they provide limited discussion or formal analysis on how the approach would extend to even higher-dimensional problems, as encountered in some real-world robotic applications). A deeper investigation into scalability limits, such as computational complexity or the effects of increasing dimensionality on solution accuracy, would strengthen this claim.

**Essential References Not Discussed:**

N/A

**Experimental Designs Or Analyses:**

The experiments primarily involve 2D environments, likely due to the assumption of known system dynamics. While these tasks are reasonable for evaluating safety constraints, incorporating higher-dimensional agents (e.g., quadrupeds like Ant from Safety Gym) would better assess scalability and applicability to more complex scenarios.

Task 2 (Pursuer-Evader) is unclear. The description refers to an "evader vehicle," but the visualizations appear to show pursuer vehicles interacting with humans. This raises several questions:

- What exactly is the task? Is the pursuer meant to collide with the evader, follow it at a safe distance, or something else?
- If the goal is to track the evader, what defines successful tracking?
- Does the pursuer ever need to prioritize obstacle avoidance over tracking?

Additionally, the environment complexity could be increased to better challenge the proposed method. Consider scenarios where:

- More obstacles block the direct path to the target.
- The free space is more constrained (e.g., narrow passages), requiring more precise maneuvering under safety constraints.

**Methods And Evaluation Criteria:**

The proposed method is well-motivated for the problem at hand. The authors reformulate the state-constrained optimal control problem in epigraph form, introducing an auxiliary value function that incorporates an additional state variable. This enables the problem to be solved using a Hamilton-Jacobi-Bellman (HJB) approach, leveraging physics-informed machine learning to approximate solutions efficiently.

**Baselines**:
The chosen benchmark tasks are reasonable but could be expanded (see "Experimental Designs or Analyses" for details).

However, the baseline selection could be improved. The current baselines include:
- C-SAC and MPPI, which are soft-constrained methods that do not enforce strict safety.
- MPPI-CBF, which introduces a hard safety constraint but is the only explicitly safe baseline.

While MPPI-CBF provides a useful comparison, a single safe baseline is insufficient to fully assess the method’s safety-performance tradeoffs. Additionally, soft-constrained baselines like C-SAC may not be the best comparison since they treat safety as a penalty rather than a hard constraint, potentially ignoring safety violations rather than actively enforcing constraint satisfaction.

To strengthen the evaluation, the authors should consider safe reinforcement learning baselines that explicitly handle hard constraints, such as: Constrained Policy Optimization (CPO), Interior-Point Policy Optimization (IPO).
These methods dynamically adjust constraint handling (e.g., by tuning Lagrange multipliers) and would provide a more rigorous safety comparison.

**Evaluation Metrics**:
The evaluation is based on cumulative cost (to measure performance), and safety rate (percentage of trajectories that remain safe). While these metrics are relevant, they are not sufficient to fully characterize the safety-performance tradeoff. Some limitations include:

- Cumulative cost does not explicitly account for constraint satisfaction trade-offs, i.e., two methods with similar costs might differ significantly in how well they satisfy safety constraints.

- Safety rate provides only a binary measure so it does not capture the severity or frequency of constraint violations (e.g., how far a trajectory enters an unsafe region).

Additional metrics could provide deeper insights, such as:
- Constraint violation magnitude: Quantifying how much and how often safety constraints are violated.

- Worst-case safety violations: To ensure the method does not fail catastrophically in rare but critical scenarios.

- Computational efficiency: Since the approach is meant to scale, measuring inference time or sample efficiency would strengthen claims about scalability.

**Other Comments Or Suggestions:**

Task 2 description can be rephrased (see my comments above).

**Other Strengths And Weaknesses:**

__Strengths__:

- The epigraph formulation effectively transforms the constrained problem into an augmented state space, facilitating tractable optimization.
- Theoretical guarantees for safety verification and performance quantification strengthen the framework’s reliability.

__Weaknesses__:

- Solving high-dimensional HJB Partial Differential Equations (PDEs) remains computationally expensive, limiting real-time applicability.

- Assumes known (or well-learned) system dynamics, restricting applicability to environments with uncertain or evolving dynamics.

- Limited comparison against strong safe control baselines, reducing the robustness of empirical validation.

- Experimental tasks are relatively simple, likely due to the dependence on known dynamics, limiting generalizability to more complex settings.

**Questions For Authors:**

- How does the learned auxiliary value function $\hat{V}$ compare to Control Barrier Functions in ensuring safety? Can it be directly used as a CBF, or how would it integrate with CBF-based methods? A response clarifying this relationship would help assess whether the approach can be leveraged in existing safety-critical control frameworks.


- How well does the proposed framework transfer to more complex, high-dimensional systems with unknown or highly nonlinear dynamics? Have you tested it beyond the benchmark tasks? Demonstrating broader applicability would strengthen the paper’s impact.


- If the assumed or learned system dynamics deviate significantly from reality, how does the approach perform in terms of both safety and optimality? Have you evaluated it under model uncertainty or system disturbances? A discussion on this would clarify the method’s robustness in real-world deployment.

**Relation To Broader Scientific Literature:**

The use of epigraph reformulation to enforce hard constraints and conformal prediction for probabilistic safety verification appears to be a novel combination that extends prior work in safe reinforcement learning. A deeper comparison with safe RL methods (e.g., Constrained Policy Optimization, Interior-Point Policy Optimization) and HJ-reachability-based safety filtering could help clarify how this approach advances the state of the art.

**Theoretical Claims:**

The paper presents several theoretical claims, primarily related to the Hamilton-Jacobi-Bellman (HJB) formulation, epigraph reformulation, and conformal prediction-based verification. While I am not deeply familiar with the theoretical aspects of this work, I have outlined some potential areas for further scrutiny. My suggestions may not be fully comprehensive.

The conformal prediction-based safety verification (Theorem 3.1) and performance quantification (Theorem 3.2) leverage results from distribution-free uncertainty quantification. These rely on assumptions about the distribution of sampled states, and their applicability to high-dimensional control problems could benefit from additional discussion.

The method learns the HJB solution using a physics-informed neural network. Theoretical guarantees on convergence or approximation error bounds for the learned solution are not explicitly discussed, which could impact scalability and accuracy in high-dimensional settings.

---

> ### Author Rebuttal · Authors · 2025-03-31
>
> # Clarification regarding the Pursuer Evader Problem
>
> In the Pursuer-Evader case study (Section 4.2), the objective is for a pursuer robot to chase a moving target and reach as close to this target as possible within the time horizon. The human figure serves only as a placeholder for the evader and can be substituted with any other entity (e.g., another robot). The key aspect of the problem is the interaction between the pursuer and the evader, both of which follow the dynamics outlined in Appendix B.2. The quality of tracking is encoded by the performance cost, which at each time step is equal to the $l_2$ distance between the evader and the pursuer. The lower the cumulative performance cost, the better the quality of tracking. In addition to tracking, the pursuer needs to avoid collisions with obstacles, which is encoded as a safety constraint. Due to this constraint, the pursuer indeed needs to prioritize obstacle avoidance over tracking at times. An example of this is shown in Figure 1 (anonymous link: https://shorturl.at/gWIrw). Here, the pursuer vehicle can achieve a lower tracking cost by cutting through the obstacle; however, due to the collision-avoidance constraint, the pursuer goes around the obstacle and intercepts the evader from behind.
>
> # Comparison with Other Safe RL Baselines
>
> In response to reviewer suggestions, we benchmarked our method against additional safe RL baselines: CPO [1], IPO [2], and PPO-Lagrangian [3] on the Pursuer-Evader problem. Our results are summarized below:
>
> | Method | Safety Rate (%) | % Cost Higher than Our Method |
> |------------------|-----------------|-------------------------------|
> |CPO|75.11| 123.14|
> |IPO|72.3 |110.34|
> |PPO-Lag|63.33|114.63|
>
> These experiments reveal that although CPO and IPO achieve marginally higher safety rates compared to C-SAC, all safe RL baselines underperform compared to the proposed approach in both safety and cost metrics.
>
> # Scalability of the Proposed Method
>
> We assess the computational scalability of the proposed approach by reporting offline and online computation times across 3 case studies:
>
> | Task | Dim | Offline Time | Online Time |
> |------------------------|----------------|--------------|-------------|
> | Boat Navigation|2|122 min|2 ms|
> | Evader Chasing|8|195 min|2 ms|
> | Multi-Agent Navigation|20|323min|2 ms|
>
> Traditional grid-based methods exhibit exponentially scaling computation complexity and become intractable for higher-dimensional problems (8D and 20D). In contrast, our method shows only a modest increase in the offline computation time when going from a 2D to a 20D system while maintaining real-time inference (2 ms) across all dimensions—a key advantage for real robotic applications.
>
> # Relations to Control Barrier Functions (CBFs)
>
> The learned auxiliary value function, $\hat{V}$, captures a notion of safety similar to CBFs. In particular, the condition $\hat{V}(x,z) \leq 0$ defines the safe region for the system, while $\hat{V}(x,z) > 0$ corresponds to unsafe states-mirroring the core idea behind CBFs. Therefore, $\hat{V}(x,z)$ can directly serve as a CBF for the system.
> However, importantly, $\hat{V}(x,z)$ extends beyond standard CBFs by also capturing performance objectives. Through the parameter $z$, one can modulate the trade-off between safety and performance, effectively generating a family of CBF-like functions with different safety-performance characteristics. As a result, $\hat{V}$ can serve as a CBF in traditional settings or be integrated into CBF-based safety frameworks to encode both safety and performance objectives.
>
> # Tackling Model Uncertainty/Disturbances
>
> We discuss the extension of the proposed approach to account for model uncertainty and disturbances in our rebuttal to reviewer BKAj. Due to space constraints, we respectfully refer the reviewer to that discussion for further details.
>
> # Transfer to Complex, High-Dimensional Systems
>
> The primary goal of this work is to lay the foundation for integrating PINNs with HJ Reachability to co-optimize safety and performance in autonomous systems. Our framework already demonstrates strong potential for high-dimensional applications, as evidenced by its success in a 20D multi-agent navigation task. Moreover, the safety constraints in our experiments are also highly complex—often consisting of disconnected, non-convex sets—highlighting the method's ability to handle intricate safety specifications. Further extension and testing of our framework on more complex systems is an exciting future research direction.
>
> # References
>
> [1] Achiam, J. et al. (2017). Constrained Policy Optimization. *ICML*.
>
> [2] Liu, Y. et al. (2020). IPO: Interior-point Policy Optimization under Constraints. *AAAI*.
>
> [3] Ray, A. et al. (2019). Benchmarking Safe Exploration in Deep Reinforcement Learning. *arXiv preprint*.
>
> [4] Bansal, S. et al. (2017). Hamilton-Jacobi Reachability: A Brief Overview and Recent Advances. *IEEE CDC*.

---

### Decision · Program_Chairs · 2025-05-01

**Decision:**

Accept (poster)

**Comment:**

The authors have presented a compelling approach to the critical challenge of jointly optimizing safety and performance in autonomous systems. The reviewers' concerns have been adequately addressed in the rebuttal. Based on the strength of the proposed framework and the authors' responses, this paper is suitable for acceptance.